# Conserved linear dynamics of single-molecule Brownian motion

Maged F. Serag[1] & Satoshi Habuchi[1]

Macromolecular diffusion in homogeneous fluid at length scales greater than the size of the molecule is regarded as a random process. The mean-squared displacement (MSD) of molecules in this regime increases linearly with time. Here we show that non-random motion of DNA molecules in this regime that is undetectable by the MSD analysis can be quantified by characterizing the molecular motion relative to a latticed frame of reference. Our lattice occupancy analysis reveals unexpected sub-modes of motion of DNA that deviate from expected random motion in the linear, diffusive regime. We demonstrate that a subtle interplay between these sub-modes causes the overall diffusive motion of DNA to appear to conform to the linear regime. Our results show that apparently random motion of macromolecules could be governed by non-random dynamics that are detectable only by their relative motion. Our analytical approach should advance broad understanding of diffusion processes of fundamental relevance.

[1] Biological and Environmental Sciences and Engineering Division, King Abdullah University of Science and Technology (KAUST), Thuwal 23955-6900, Saudi Arabia. Correspondence and requests for materials should be addressed to M.F.S (email: maged.serag@kaust.edu.sa) or to S.H. (email: satoshi.habuchi@kaust.edu.sa).

Brownian motion, as famously explained by Albert Einstein in 1905, is a process during which tiny particles move randomly in a homogeneous isotropic fluid as they experience independent molecular collisions from the thermally excited fluid molecules[1]. This erratic particle motion has fascinated scientists for most of the last two centuries. Probing biomolecular interactions[2], imaging of cell organelles and nanostructures in three dimensions[3,4] and building molecular motors[5] are among the major scientific applications of Brownian motion. The full molecular-scale context of Brownian motion occurs in three basic regimes. In the first regime before any molecular collisions, the particle shows a ballistic-like motion[6]. In the second regime, which begins when the particle interacts with fluid molecules and the resulting friction creates local vortices that act on the particle, the molecular motion is affected by hydrodynamic forces of the fluid[7]. These ballistic and hydrodynamic regimes, therefore, deviate from the random Brownian motion. In the third regime during which the particle diffuses its own radius, statistically independent collisions dominate the motion, causing the overall motion of the particle to be random. The random Brownian motion causes the random positions of the particle, and therefore, the mean-squared displacement (MSD) of molecules increases linearly with time.

Among the dynamics of particles in fluids[6–8], those of DNA are rather unique[9–11]. DNA is a semiflexible polymer; its motion in a homogeneous isotropic medium and within its radius of gyration is governed by the constraints imposed by its chain connectivity and by various intramolecular hydrodynamic interactions[10,11]. These dynamics, which complicate the hydrodynamic regime of DNA molecules, often cross over and partially affect their linear diffusive regime[12,13]. Consequently, at diffusion distances close to the radius of gyration, the motion of DNA molecules is non-linear and subdiffusive. At longer diffusion distances and at long time scales, DNA molecules diffuse, following the expected behaviour of a polymer as a whole, and MSD is linear with time[12,13]. Although slow conformational fluctuations have been observed within the time scale of the diffusive motion of DNA[14], the physical origin of these fluctuations and whether they affect the diffusive motion of DNA have not been determined. This fundamental understanding has thus far been hampered by a lack of both theory and analytical tools that give access to the diffusive regime of macromolecules.

Here we report the development of new theoretical framework and analytical tool that can capture the motion of macromolecules in their diffusive regime by characterizing the motion relative to a latticed frame of reference. Our new method—lattice occupancy analysis—reveals unexpected sub-modes of motion of DNA molecules that deviate from the expected random motion in the linear, diffusive regime.

## Results

**Theoretical framework of lattice occupancy analysis.** Brownian motion is typically viewed in terms of the absolute positions of single molecules, which are the hallmark of MSD analysis[6,7,9,10,15–17]. In this study, we consider their linear, diffusive regime not in terms of MSD's absolute measurement but in terms of a relative measurement. In this measurement, we study the motion of single molecules with respect to a virtual latticed frame of reference with which we analyse how often the molecule steps into new lattice sites in a diffusion space during its motion (Fig. 1a, Supplementary Data 1). The experimental probability of lattice occupancy ($P_t$) at time $t$ is given by

$$P_t = \frac{<k_t>}{n}, \qquad (1)$$

where $n$ is the number of steps and $<k_t>$ is the average number of visits to new lattice sites. According to one-dimensional (1D) random diffusion theory, the probability distribution ($p$) of finding the particle at different lattice sites, $q$, is described by[18,19]

$$p = \frac{1}{\sqrt{2\pi n l^2}} \exp\left(\frac{-q^2}{2nl^2}\right) = \frac{1}{\sigma\sqrt{2\pi}} \exp\left(\frac{-q^2}{2\sigma^2}\right), \qquad (2)$$

where $n$, $l$ and $\sigma$ respectively denote the number of step(s), the step size and the s.d. of $p$. For simplicity, we set $l = m$, where $m$ is the side length of the lattice. The particle executes $n$ steps in time $t = n\tau$, where $\tau$ is a unit of time. Since the diffusive spreading[20] of the particle (that is, spreading of the diffusing particles in the space away from their initial position) is a measure of the rate at which the particle spreads out in space during time $t$[19], it is expressed as the increase in $\sigma$ as the square root of $t$ increases (Fig. 1b)[20]. The rate of diffusive spreading in the lattice's 1D space (that is, the number of visits to new lattice sites) is equivalent to the probability of occurrence of visits to new lattice sites at time $t$ ($P_t$). $P_t$ is expressed as

$$P_t = \frac{1}{l\sqrt{2\pi}} t^{-0.5}. \qquad (3)$$

$P_t$ obtained from a simulated 1D random diffusion trajectory agrees well with equation (3) (Supplementary Fig. 1). In two-dimensional (2D) random diffusion, the new position of the molecule depends on the step size, the direction of the motion and where the last step ends. Successive steps could therefore occur in the same lattice site, and the rate of the power law decay (that is, the rate of visiting new lattice sites in 2D) could thus accordingly decreases. Thus, $P_t$ in 2D space ($P_t^{2D}$) can be given by

$$P_t^{2D} = \frac{1}{l\sqrt{2\pi}} t^{-\beta}, \qquad (4)$$

where $\beta$ is the scaling exponent of the power-law decay. $P_t^{2D}$ obtained from a simulated 2D random diffusion trajectory agrees well with equation (4) (Fig. 1c). Our lattice occupancy analysis of simulated directed and confined 2D motions showed that the values of $\beta$ were dependent on the diffusion mode (Supplementary Fig. 2).

**Diffusive motion of DNA in a latticed 2D space.** Based on this theoretical framework of molecular motion, we characterized the spatiotemporal pattern of the diffusive motion of DNA (linear ColE$_1$ DNA (ref. 21)) using single-molecule fluorescence microscopy (Fig. 2a,b and Supplementary Video 1, see Supplementary Note 1). Standard MSD analysis of the diffusion trajectories (Fig. 2b) showed sub-diffusive dynamics of DNA at length scales close to its radius of gyration ($R_g = 0.186 \mu m$ (50 ms))[21] (Fig. 2c). This regime arises due to the crossover of the hydrodynamic regime of DNA[10] (shown in red in Fig. 2c). The MSD-time-lag ($\Delta t$) profile at length scales larger than the radius of gyration of DNA, that is, at a time scale longer than 50 ms, reflected the linear, diffusive regime of its Brownian motion (shown in blue in Fig. 2c and Supplementary Fig. 3a). On the other hand, the MSD of spherical polymer nanospheres exhibited a linear increase with time at all scales (Supplementary Fig. 3b). This indicated the pure random walk of these nanospheres; we therefore used them as a control throughout the study (Supplementary Fig. 3b and Supplementary Video 2). To characterize the diffusive motion of the DNA and nanospheres in the linear, diffusive regime with respect to the latticed frame, we calculated a temporal profile of $P_{25}$ (the probability of new visits in a time lag of $25\Delta t$, Fig. 1c). $P_{25}$ at each time point was determined by applying a $50\Delta t$ time window. The temporal profile of $P_{25}$ was then obtained by sliding the time window along the trajectory (Fig. 2b,d). The time-dependent $P_{25}$ values exhibited fluctuations between a high lattice occupancy

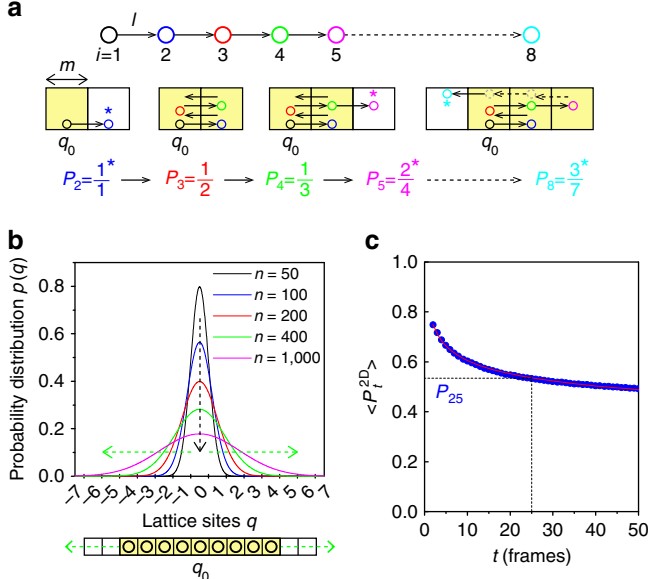

**Figure 1 | Theoretical framework of lattice occupancy analysis.**
(**a**) Schematic diagram illustrating the motion of a single particle on 1D lattice frame. The motion starts at lattice site $q_0$ of side length $m$ where the particle moves either to the right or to the left with step size $l = m$. The probability of occurrence of visits to new lattice sites at time $t$ ($P_t$) decreases as $t$ increases. (**b**) Probability distribution ($p$) of finding a particle at different lattice sites after $n$ steps ($n = t/\tau$) calculated using 1D random diffusion theory (equation (2)). The s.d.'s increase (green dashed lines), whereas the peak heights decrease (black dashed lines) with the square root of $n$ (equation (2)) (**c**) Average probability of occurrence of visits to new lattice sites at time $t$ ($<P_t^{2D}>$) obtained from 100 simulated 2D random diffusion trajectories. The lattice size ($m$) was set to 160 nm. The step sizes of the trajectories were generated using equation (5) ($r = 160$ nm). The red line shows the fitting to equation (4). The scaling exponent ($\beta$) obtained by the fitting is 0.133 for the random walk.

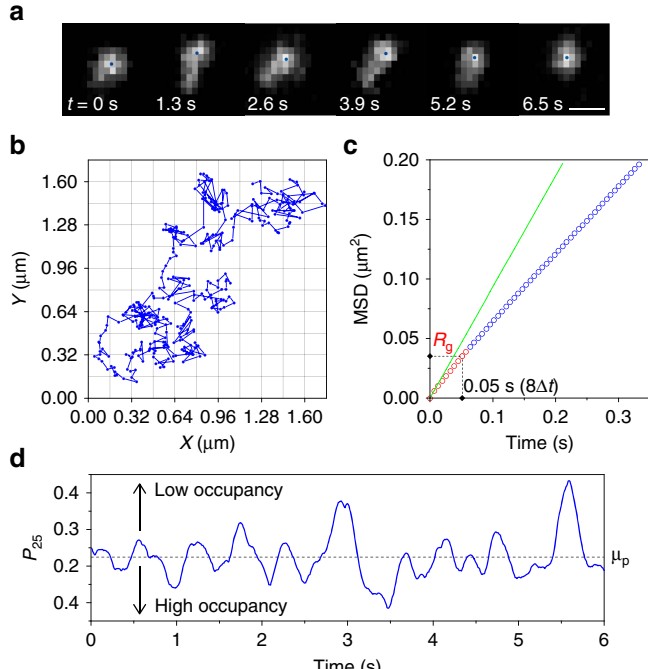

**Figure 2 | Single-molecule tracking and analyses of $ColE_1$ DNA.** (**a**) Time lapse fluorescence images of a single DNA molecule. The blue circles show the 2D positions of the molecule determined by the tracking. Scale bar = 1 μm. (**b**) Diffusion trajectory of $ColE_1$ DNA mapped onto the latticed 2D space with 0.16 μm side length of the lattice (See Supplementary Note 1). (**c**) MSD-$\Delta t$ profile of $ColE_1$ DNA ($\Delta t = 6.4$ ms). $R_g$ shows the radius of gyration of $ColE_1$ DNA. The red and blue circles highlight the parts of the MSD-$\Delta t$ profile that show sub-diffusive and linear-diffusive regimes. The green line is the theoretical MSD-$\Delta t$ profile. (**d**) Temporal $P_{25}$ profile of $ColE_1$ DNA. $\mu_p$ shows the mean $P_{25}$ value.

mode (low $P_{25}$ value or a few visits to new lattice sites) and a low lattice occupancy mode (high $P_{25}$ value or more visits to new lattice sites). We collectively refer to the modes that result from the lattice occupancy analysis as relative modes (Fig. 2d).

**Analysis of the hidden non-random diffusive dynamics of DNA.**
Next, we analysed the diffusive dynamics of DNA based on the time-dependent $P_{25}$ profile. Because Brownian motion is fractal in nature, its temporal fluctuations are random at all scales. Any fluctuations, including $P_{25}$, are hence invariant regardless of the time scale used to probe the motion[22]. On the other hand, non-random motion occurring in the linear, diffusive regime causes time-scale-dependent fluctuations. Such fluctuations can be captured by using detrended fluctuation analysis (DFA) and by calculation of the Hurst exponent (HE) (Supplementary Data 2, see Methods for the details)[23]. The time-scale-dependent fluctuations due to the non-random motion results in larger HE (HE > 0.5) compared with that obtained from random motion (HE = 0.5).

To provide statistically robust HE estimates, we joined 98 single-molecule tracks end-to-end and generated a long probability time series (Fig. 3a, Supplementary Fig. 4). Any systematic errors that could arise from the end-to-end connections were evaluated by calculating the HE of 100 shuffled replicates by randomizing the order of the connections between the original trajectories (Supplementary Fig. 4a). We then compared the HE of these experimental replicates with those of

simulated replicates to identify any deviations from random behaviour, if any, and also to identify the physical origin of these deviations. The simulated trajectories were generated by randomizing the order of both the step sizes (S) and the step directions (angles (A)) of the original experimental replicates (denoted as $S_rA_r$ simulated replicates), by randomizing the angles while maintaining the order of the step sizes ($S_iA_r$) or by randomizing the step sizes while maintaining the order of the angles ($S_rA_i$). The analyses uncovered a dramatic positive shift of the calculated HE profiles of experimental replicates of DNA (Fig. 3c blue lines, Fig. 3e blue line, Fig. 3g blue line), but not of the nanospheres (Fig. 3b,d,f, Supplementary Fig. 5), the simulated replicates of $S_rA_r$ (Fig. 3e green lines, Fig. 3g green line), and the simulated replicates of $S_rA_i$ (Fig. 3e black lines, Fig. 3g black line). The HE profiles of the experimental replicates of DNA (Fig. 3c blue lines) exhibit this positive shift compared with those of the simulated replicates of $S_rA_r$ (Fig. 3c green lines) even at $10\Delta t$ at which the MSD-$10\Delta t$ profile exhibits linear behaviour (Supplementary Fig. 3a). The results clearly demonstrate that the non-random motion of DNA in the linear, diffusion regime, which is not captured by MSD analysis, is revealed by lattice occupancy analysis.

Interestingly, the HE profile of the simulated replicates of $S_iA_r$ (Fig. 3e red lines, Fig. 3g red line) partially overlaps that of the experimental replicates, indicating that the HE shift is unique to DNA motion and it is partially related to the order of the step sizes but not to the order of the angles. Furthermore, MSD analyses of the $S_rA_i$ replicates (Fig. 3h black line) produced a sub-diffusive profile, whereas MSD analyses of the $S_rA_r$ (Fig. 3h

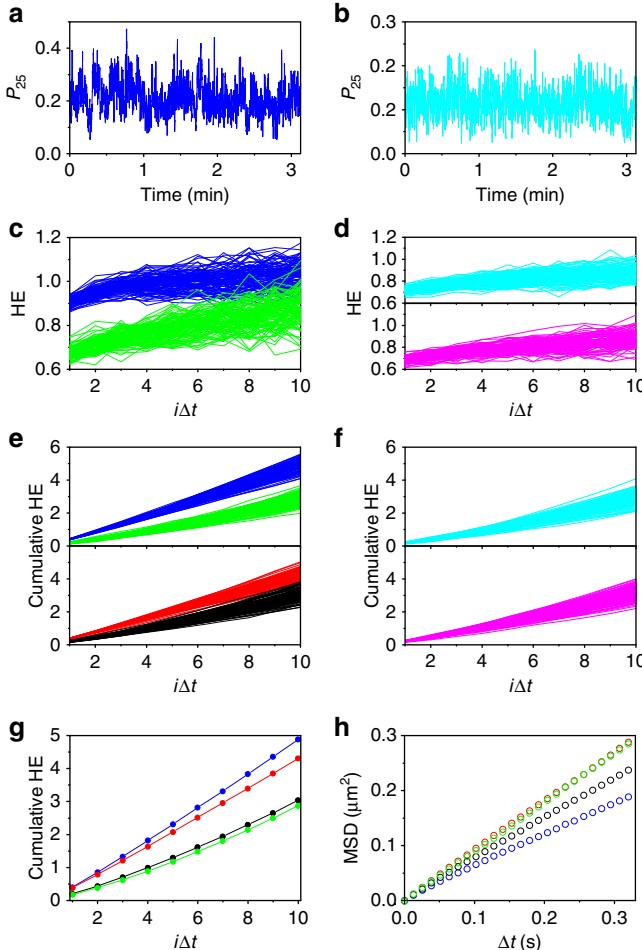

**Figure 3 | Detrending fluctuation analyses of the time-dependent $P_{25}$ profile of ColE$_1$ DNA.** (**a**) Time-dependent $P_{25}$ profile of the DNA molecules. (**b**) Time-dependent $P_{25}$ profile of the nanospheres. (**c**) Hurst exponent (HE)-$i\Delta t$ profiles of the experimental (blue) and the $S_rA_r$ replicates (green) of the ColE$_1$ DNA. (**d**) HE-$i\Delta t$ profiles of the experimental (cyan) and the $S_rA_r$ replicates (magenta) of the nanospheres. (**e**) Cumulative HE of the experimental (blue), the $S_rA_r$ (green), the $S_iA_r$ (red) and the $S_rA_i$ (black) replicates of the ColE$_1$ DNA. (**f**) Cumulative HE of the experimental (cyan) and the $S_rA_r$ replicates (magenta) of the nanospheres. (**g**) Averaged cumulative HE of the experimental and simulated replicates shown in **e**. The same colour coding as in **e** is used. (**h**) MSD-$\Delta t$ profiles of the experimental and the simulated replicates. The same colour coding as in **e** is used.

green line) and the $S_iA_r$ (Fig. 3h red line) replicates produced linear profiles. These results indicate that the sub-diffusive behaviour of DNA observed in the hydrodynamic non-random regime (Fig. 2c) results from the order of the directions of the steps of the DNA walk.

**Identification of the diffusion modes of DNA.** Next, we investigated the origin of the hidden non-random motion of DNA molecules in the linear, diffusive regime as revealed by lattice occupancy analysis. According to the random walk theory, statistical variations in step sizes and step directions yield trajectories that resemble by chance those of directed and confined modes of diffusion (Supplementary Fig. 6a–f)[24,25]. The MSD-$\Delta t$ profiles obtained from the trajectories with directed-like and confined-like modes of motion thus respectively exhibited concave and convex curves as described by equations

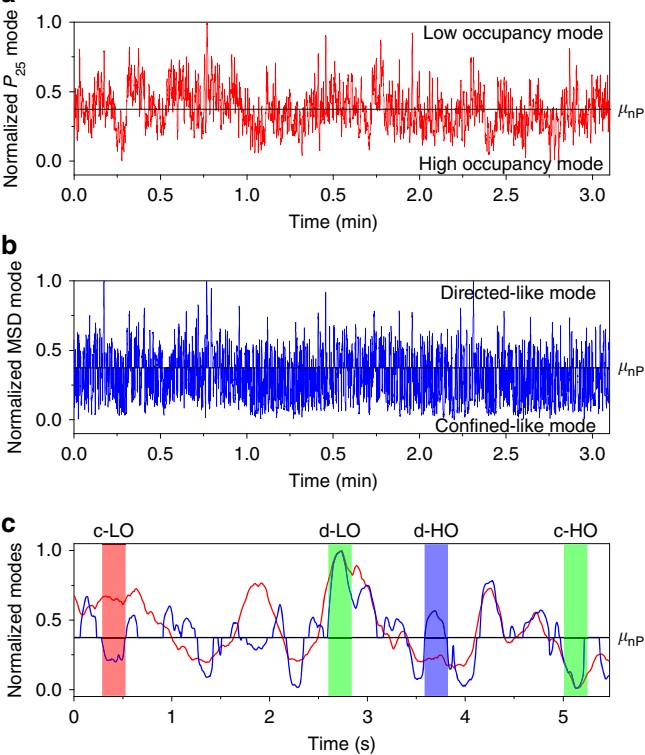

**Figure 4 | Comparison of the MSD-$\Delta t$ and $P_{25}$ temporal profiles.** (**a**) Normalized $P_{25}$ temporal profile of an experimental replicate of ColE$_1$ DNA. $\mu_{np}$ is the mean of the normalized $P_{25}$ profile. (**b**) Normalized temporal profile of the absolute diffusion modes (directed-like and confined-like) of ColE$_1$ DNA obtained from the MSD-$\Delta t$ plots of an experimental replicate at each time point. (**c**) Enlarged view of the superimposed normalized $P_{25}$ (red) and MSD-$\Delta t$ (blue) temporal profiles. The green shadings highlight the d-LO and c-HO sub-modes. The red and blue shadings highlight the c-LO and d-HO sub-modes, respectively.

(10) and (11) (see Methods; Supplementary Fig. 6g,h). These apparent deviations from random motion arise from the limited length of the experimental trajectories. Because these apparent deviations are viewed as parts of the random fractal nature of the diffusive regime of Brownian motion, they are persistent at all time scales[22,24,25]. Thus, the temporal fluctuations of the $P_{25}$ value occurring at the time scale of the linear, diffusive regime are accounted for by both the non-random motion of DNA and the apparent deviations (directed-like and confined-like modes) from the random motion. Indeed, lattice occupancy analyses of simulated trajectories displaying directed-like and confined-like modes of diffusion respectively yield a low lattice occupancy mode (high $P_{25}$ value) and a high lattice occupancy mode (low $P_{25}$ value) (Supplementary Fig. 2).

As a first step in distinguishing the apparent non-random diffusion caused by statistical variations intrinsic to Brownian motion and actual non-random diffusion of DNA, we compared the temporal profiles retrieved from the relevant analytical tool in each case. Specifically, we compared the temporal behaviour characterized by MSD analysis (apparent non-random diffusion) and the temporal $P_{25}$ profile (actual non-random diffusion). We first normalized the temporal profile between 0 and 1 and split the trajectory into high and low lattice occupancy modes at the mean ($\mu_{nP}$) (Fig. 4a, Supplementary Data 3). The characterization of the temporal behaviour by MSD analysis was conducted by first calculating the MSD-$\Delta t$ plots at each time point by applying a sliding window with a $50\Delta t$ time width (see Methods for

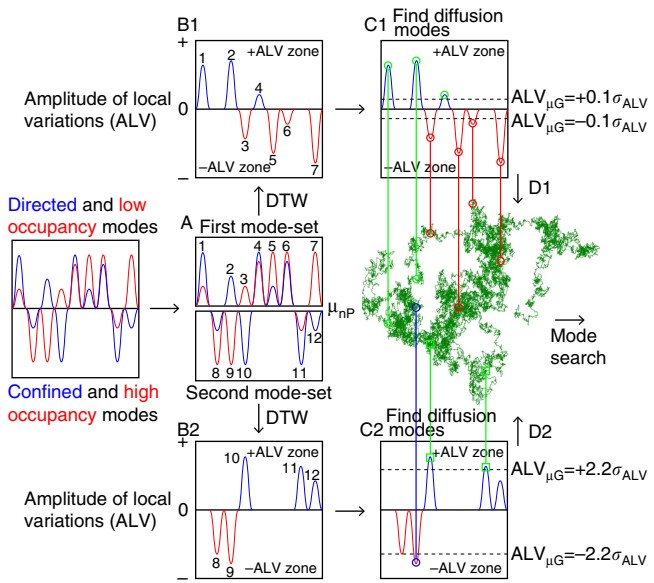

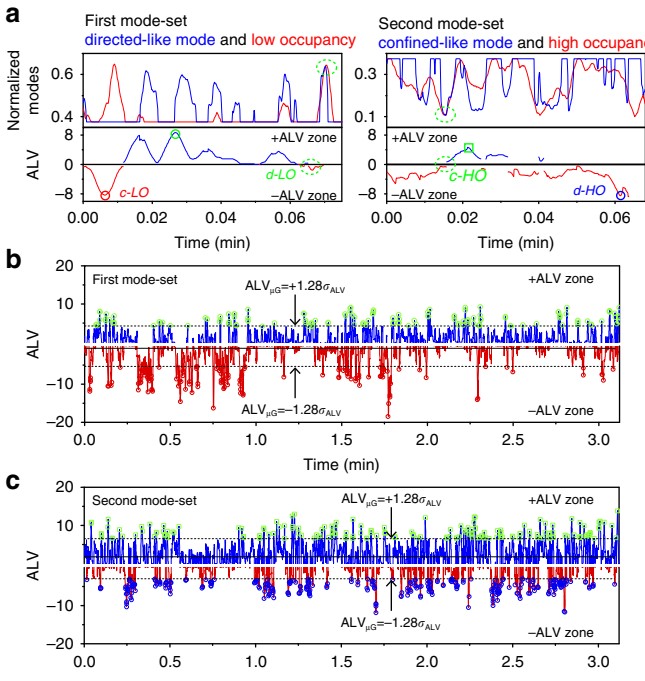

**Figure 5 | Analysis of the sub-modes of diffusion.** Schematic illustration of the four-step analyses of the sub-modes of diffusion.

**Figure 6 | Identification of the sub-modes by amplitudes of the local variations.** (**a**) Temporal profiles of normalized $P_{25}$ (red) and MSD-$\Delta t$ (blue) (top) and the amplitudes of the local variations (ALV) (bottom) of the first (left) and second (right) mode-set obtained from an experimental replicate of ColE$_1$ DNA. (**b**) ALV-$\Delta t$ profile obtained from the first mode-set. The green circles indicate that the amplitudes of the directed-like mode are higher than those of the relative mode. The red circles indicate that the amplitudes of the low occupancy mode are higher than those of the absolute mode. (**c**) ALV-$\Delta t$ profile obtained from the second mode-set. The green squares indicate that the amplitudes of the confined-like mode are lower than those of the relative mode. The blue circles indicate that the amplitudes of the high occupancy mode are lower than those of the absolute mode.

details). The directed- and confined-like motions were quantified by drift velocity ($v$, equation (10)) and the length of the confined area ($L$, equation (11)) and normalized to $\mu_{nP}$ and 1 and $\mu_{nP}$ and 0, respectively (Fig. 4b, Supplementary Data 4). We refer to these directed-like and confined-like modes that result from MSD's absolute measurement as absolute modes. We then superimposed the normalized $P_{25}$ and MSD (that is, $v$ and $L$) temporal profiles for analysing the modes of diffusion qualitatively (Fig. 4c). Comparison of the two temporal profiles identified the time instances at which the correlation between the $P_{25}$ and MSD profiles was positive (directed-like motion with low lattice occupancy (high $P_{25}$) mode ($d$-LO sub-mode) and confined-like motion with high lattice occupancy (low $P_{25}$) mode ($c$-HO sub-mode)) (Fig. 4c). However, the temporal profiles also revealed that they are not always positively correlated with each other (Fig. 4c). Specifically, the temporal profiles showed that the confined-like motion was sometime correlated with the low lattice occupancy mode ($c$-LO sub-mode) and that the directed-like mode was sometime correlated with the high lattice occupancy mode ($d$-HO sub-mode) (Fig. 4c).

**Characterization of the non-random diffusion modes of DNA.** To determine if any of these four sub-modes ($d$-LO, $c$-HO, $c$-LO and $d$-HO) causes the non-random motion of DNA, we devised a four-step analytical approach (Fig. 5 and Supplementary Note 2). First, we split the normalized $P_{25}$ and MSD temporal profiles (Fig. 4a,b) into two mode-sets at $\mu_{nP}$ (step A in Fig. 5, Supplementary Note 3) and calculated the amplitudes of the local variations (ALV) between the two profiles (step B in Fig. 5, Fig. 6a) using the dynamic time warping (DTW) algorithm (Supplementary Data 5, see Methods for details) to quantify the deviations between the two time profiles. The DTW algorithm provided a negative ALV ($-$ALV zone) when the amplitude in the $P_{25}$ profile was larger than the amplitude in the MSD profile. On the other hand, the DTW algorithm provided a positive ALV ($+$ALV zone) when the amplitude in the $P_{25}$ profile was smaller than the amplitude in the MSD profile. Thus, extremely negative ALV values in the first and second mode-sets reflect the $c$-LO and $d$-HO sub-modes, respectively, and extremely positive ALV values in the first and second mode-sets reflect the $d$-HO and $c$-LO modes, respectively (Fig. 6a). We then used the ALV values

above or below the thresholds to distinguish between different sub-modes that exist in the experimental replicates (step C in Fig. 5 and Fig. 6b,c). We used the thresholds defined by the mean (ALV$_{\mu G}$) and the s.d. of the ALV profiles ($\sigma_{ALV}$, between $2.2\sigma_{ALV}$ (1.4 % of the data) and $0.1\sigma_{ALV}$ (46% of the data)), which were calculated from the 100 experimental replicates. We next extracted the diffusion sub-trajectories corresponding to these sub-modes. Using these sub-trajectories, we calculated MSD-$\Delta t$ plots and step-size distributions to analyse the diffusion modes (step D in Fig. 5). Finally, we compared these trajectories with their respective trajectories obtained from the nanospheres and the simulated $S_rA_r$ replicates so that we could discern whether or not the experimental sub-modes exhibited non-random behaviours.

Figure 6a–c shows that the deviations between the two profiles are easily identified by calculating ALV using the DTW algorithm. We first examined the step-size distributions of the extracted diffusion sub-trajectories whose time regions are defined by setting the ALV threshold to $\pm 2.2\sigma_{ALV}$ (Fig. 7a, Supplementary Data 6). While the distributions do not display any deviations from the 2D random diffusion theory[26], the distributions obtained from the $-$ALV zones (Fig. 7a top) exhibit either larger (first mode-set) or smaller (second mode-set) mean step sizes compared with those obtained from the $+$ALV zones (Fig. 7a bottom). The mean step sizes in the $-$ALV zones clearly show the dependency on the threshold level, demonstrating that

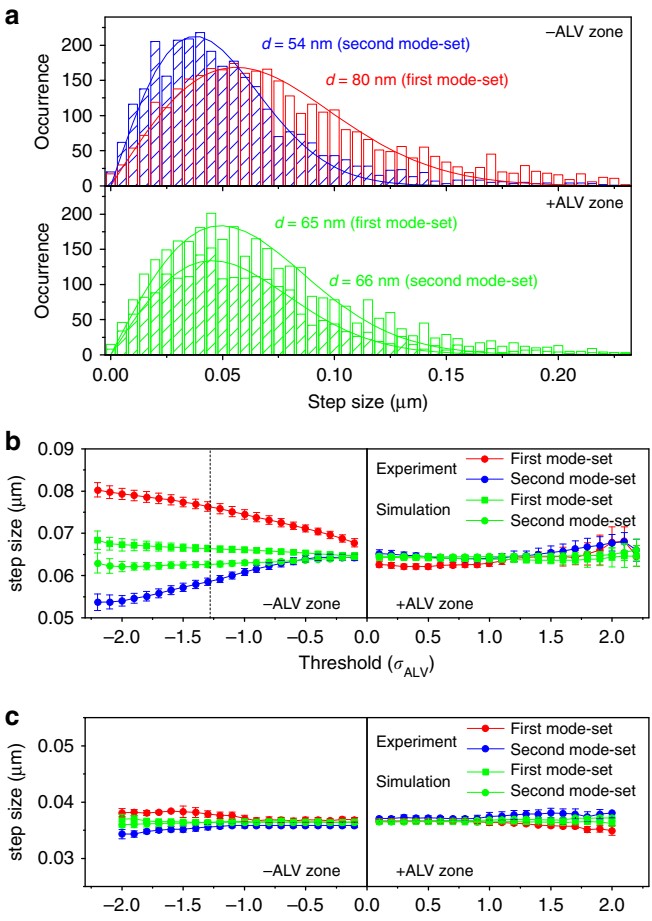

**Figure 7 | Step sizes in the sub-trajectories.** (**a**) Frequency histograms of the step-size distributions of the sub-trajectories obtained from an experimental replicate of ColE$_1$ DNA whose temporal positions are extracted from ALV (red and blue histograms for the first and second mode-set in −ALV zone, and green and shaded-green histograms for the first and second mode-set in + ALV zone) by setting the thresholds at ± 2.2$\sigma_{ALV}$. (**b**) The mean step sizes obtained from the experimental (red and blue circles for the first and second mode-set, respectively) and S$_r$A$_r$ simulated (green circles and green squares for the second and first mode-set, respectively) sub-trajectories of ColE$_1$ DNA at different threshold levels (0.1$\sigma_{ALV}$ – 2.2$\sigma_{ALV}$). The mean step-sizes were determined by fitting the step-size distributions to equation (5). The dashed line shows the ALV threshold at which 10% of the data are collected. (**c**) The mean step sizes obtained from the experimental (red and blue circles for the first and second mode-set, respectively) and S$_r$A$_r$ simulated (green circles and green squares for the second and first mode-set, respectively) sub-trajectories of the nanospheres at different threshold levels (0.1$\sigma_{ALV}$ − 2.0$\sigma_{ALV}$). Error bars in **b**,**c** correspond to the s.d.'s of the step sizes at each threshold level obtained from 100 replicates.

the negative peaks in the ALV plots are responsible for the larger and smaller step sizes in the first and the second mode-set, respectively (Fig. 7b). We did not observe this threshold dependency in the negative peaks detected in the ALV plots obtained from the nanospheres (Fig. 7c) and the simulated S$_r$A$_r$ replicates (Fig. 7b) (that is, those peaks are assigned to the apparent non-random diffusion caused by the statistical variations intrinsic to Brownian motion). These results further demonstrate that the sub-trajectories corresponding to the ALV peaks in the negative zones display non-random diffusion modes. Since the negative ALV peaks correspond to the larger amplitudes

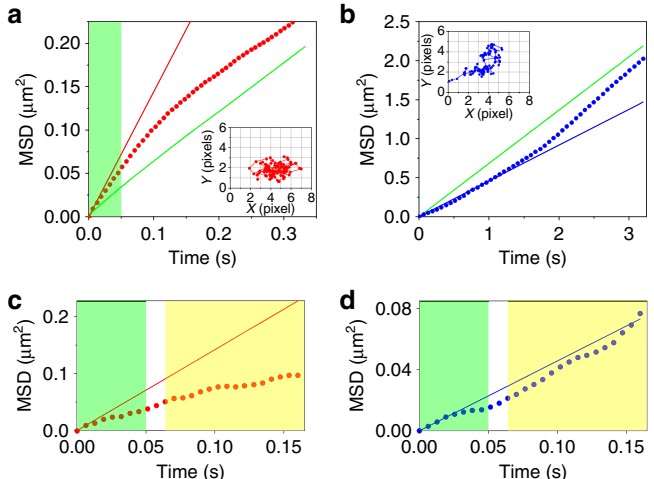

**Figure 8 | MSD analyses of the sub-trajectories.** (**a**) Averaged MSD-$\Delta t$ profile of the sub-trajectories captured in the −ALV zone of the first mode-set. The green line shows the overall MSD-$\Delta t$ profile obtained from the original experimental replicates (Fig. 2c). The red line shows the theoretical MSD-$\Delta t$ profile of the averaged MSD-$\Delta t$ profile. (**b**) Averaged MSD-10$\Delta t$ profile of the sub-trajectories captured in the −ALV zone of the second mode-set. The green line shows the overall MSD-10$\Delta t$ profile obtained from the original experimental replicates. The blue line shows the theoretical MSD-10$\Delta t$ profile of the averaged MSD-10$\Delta t$ profile. Insets in **a**,**b** show examples of single-molecule sub-trajectories obtained from the −ALV zone of the first mode-set and the −ALV zone of the second mode-set, respectively. (**c**) MSD-$\Delta t$ profile of the sub-trajectory shown in the inset of **a**. The red line shows the theoretical MSD-$\Delta t$ profile. (**d**) MSD-$\Delta t$ profile of the sub-trajectory shown in the inset of **b**. The blue line shows the theoretical MSD-$\Delta t$ profile. The green shaded area in **a**,**c**,**d** highlights part of the MSD profiles whose time scale shows sub-diffusive behaviour. The yellow shaded area in **c**,**d** highlight part of the MSD profile whose time scale shows linear diffusive behaviour. This part of the profile shows either confined-like (**c**) or directed-like motion (**d**) compared with the theoretical profile.

of the $P_{25}$ profile compared with those of the MSD profile (that is, in our analytical approach, the diffusive motion is mainly characterized by the relative diffusion modes—c-LO and d-HO sub-modes for the first and second mode-sets, respectively), the results also demonstrate that lattice occupancy analysis can capture non-random diffusion modes. The step sizes obtained from the peaks detected in the + ALV zones, which mainly reflect the absolute diffusion modes (that is, directed- or confined-like motion), do not show any deviations from the average step size (Fig. 7a–c), demonstrating that these sub-modes captured by MSD analysis do not exhibit non-random behaviour.

We further characterized the LO and HO modes detected in the above analyses by reconstructing corresponding MSD-$\Delta t$ plots (Supplementary Data 6). The MSD-$\Delta t$ plots reconstructed from the region of the diffusion trajectories that display the LO sub-mode clearly showed confined-like motion (c-LO sub-mode) (Fig. 8a,c). On the other hand, the MSD-$\Delta t$ plots obtained from the HO sub-mode region exhibited directed-like motion (d-HO sub-mode) (Fig. 8b,d). The MSD-$\Delta t$ plots of the simulated S$_r$A$_r$ replicates that were reconstructed from the regions with negative ALV peaks exhibited normal diffusion (Supplementary Fig. 7a), confirming the existence of unexpected c-LO and d-HO sub-modes that are responsible for the non-random motion of the DNA in homogeneous isotropic environments. On the other hand, the MSD-$\Delta t$ plots obtained from the positive ALV zones of the DNA trajectories were indistinguishable from the simulated

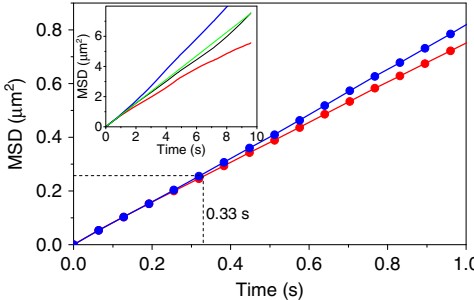

**Figure 9 | Effect of the temporal order of the step sizes on MSD-$\Delta t$ profile.** MSD-10$\Delta t$ profiles of the $c_{sh}$-$LO$ (red) and the $d_{lg}$-$HO$ (blue) simulated trajectories. The inset shows the full MSD-10$\Delta t$ profiles of these simulated trajectories. The black and green lines show the MSD profile of the original experimental replicate and its theoretical 10$\Delta t$ profile.

$S_r A_r$ replicates (Supplementary Fig. 7b). This result together with the ALV threshold-dependent step sizes (Fig. 7) confirms that the sub-modes characterized by the MSD analysis exhibit random behaviour.

To investigate the effect of the temporal order of the step sizes on the non-random motion, we replaced the larger steps of the c-LO sub-mode (Fig. 7a red) in the experimental diffusion trajectory with the randomly ordered smaller steps obtained from other modes (Fig. 7a blue and green) (denoted $c_{sh}$-LO) (Supplementary Fig. 8). The MSD-$\Delta t$ plot obtained from the simulated $c_{sh}$-LO trajectory showed a deviation from the linear MSD-$\Delta t$ profile of the original experimental replicate towards confined-like motion (Fig. 9 red), suggesting that the shorter step sizes in the time regions of the c-LO sub-mode—instead of long step sizes—caused this deviation towards confined-like motion (Supplementary Data 7). On the other hand, the MSD-$\Delta t$ plot obtained from the simulated trajectory whose short steps in the d-HO sub-mode were replaced by the randomly ordered longer steps obtained from other modes ($d_{lg}$-HO trajectory) displayed the opposite behaviour (that is, a shift towards directed-like motion, Fig. 9 blue), suggesting that the larger step-sizes in the d-HO sub-mode caused the deviation towards the directed-like motion (Supplementary Data 7). These findings further demonstrate that the non-random temporal order of the step sizes causes the non-random motion of the DNA and is consistent with c-LO and d-HO sub-modes. A characteristic time scale of the c-LO sub-mode ($\tau_{c\text{-}LO}$) was estimated to be $\tau_{c\text{-}LO} = 0.33 \pm 0.016$ s by Fourier transform analysis (Supplementary Figs 9,10). Interestingly, the $\tau_{c\text{-}LO}$ is in good agreement with the conformational relaxation time ($\tau_R = 0.34$ s) of DNA (Supplementary Fig. 11), indicating possible involvement of the conformational relaxation of DNA in non-random motion (see Discussion for the detail).

**Motion of DNA in the crossover regime**. We then examined whether or not the relative c-LO and d-HO sub-modes exist in other regimes of molecular motion. To that end, we analysed diffusion trajectories of lambda phage DNA (48,500 kbp, $R_g = 0.7\,\mu m$, Fig. 4a)[27]. Because the radius of gyration of lambda DNA is much larger than that of ColE$_1$ DNA, lambda DNA displayed a sub-diffusive MSD-$\Delta t$ profile (Fig. 10a) in the time scale that is compatible with lattice occupancy analysis. This indicates that we capture the motion of lambda DNA in its crossover regime by lattice occupancy analysis. In contrast to ColE$_1$ DNA molecules, we did not find a significant difference between the calculated HE of the experimental replicates and their $S_r A_r$ replicates (Fig. 10b, Supplementary Fig. 12a). Furthermore, we captured neither the c-LO/d-HO sub-modes (Fig. 10c) nor the non-random temporal order of the step sizes that were observed in ColE$_1$ DNA

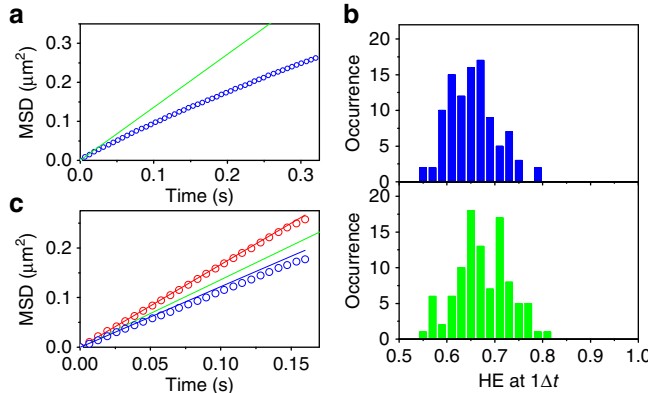

**Figure 10 | Lattice occupancy analysis of lambda DNA. (a)** MSD-$\Delta t$ profile of lambda DNA ($\Delta t = 6.4$ ms). The red line shows the theoretical MSD-$\Delta t$ profile. **(b)** Frequency histogram of the HE-1$\Delta t$ distribution of the experimental (top) and the $S_r A_r$ simulated replicates (bottom) of lambda DNA. **(c)** Averaged MSD-$\Delta t$ profiles of the sub-trajectories captured in the same way as in Fig. 7a (red) and Fig. 7b (blue). The red and blue lines are the theoretical MSD-$\Delta t$ profiles. The green line shows the overall, theoretical MSD-$\Delta t$ profile of the experimental replicates.

(Supplementary Fig. 12b). The results demonstrate that the non-random motion of DNA we captured using lattice occupancy analysis (that is, lattice occupancy modes) is observed characteristically in the linear, diffusive regime. The non-random motion of DNA in its crossover regime is better characterized by MSD analysis. Since the diffusion coefficients of ColE$_1$ and lambda DNA in our experiments were close to each other, these findings also serve as an important confirmation that the three-dimensional (3D) motion of the molecules does not affect lattice occupancy analysis.

**Discussion**

Studying the relative motion of single molecules provides a means to extract essential information on non-random dynamics that has remained inaccessible via conventional theories of absolute measurements (Supplementary Fig. 13). We used the motion of DNA relative to a square lattice to unlock a subtle dynamic regime in the Brownian motion of DNA and to uncover that diffusion speed of polymer molecules and the mode of motion have unexpected effects on 2D lattice occupancy (that is, the presence of unexpected c-LO and d-HO sub-modes). Therefore, our analytical approach is different from, yet complementary to, other analytical methods such as cumulative area tracking (CA tracking)[16,17,28]. The CA tracking method analyses the diffusion constant of single molecules by relating the mean cumulative area difference to the elapsed time. CA tracking has two major advantages. First, in contrast to our lattice occupancy analysis, which uses single-molecule localization algorithms to determine the position of the molecule, CA tracking circumvents the localization step of the molecule while employing simple tracking of a limited number of pixels (proxy pixels) that define the position of the molecule. Second, by controlling the number of proxy pixels, conformational dynamics can be simultaneously analysed by CA tracking. A major difference between our lattice occupancy algorithm and CA tracking is that the latter cannot be used to calculate 2D lattice occupancy because of the random shape of the proxy pixels. Although the effect of a change in the shape of the proxy pixels is averaged out during the calculation of the diffusion constant, a large error can be introduced in the time-dependent lattice occupancy profile. Lattice occupancy analysis is therefore complementary to CA tracking because it

allows us to characterize the relative motion of DNA and to correlate it with the conformational relaxation time of the molecule.

By using lattice occupancy and MSD analyses, we found that the relative motion of DNA is dramatically affected by the distribution of its step sizes. Specifically, an increase in the step sizes during confined-like motion pushes the molecular positions apart, and the relative motion thus exhibits low lattice occupancy (c-LO sub-modes). Conversely, a decrease in the step sizes during directed-like motion causes the relative motion to exhibit high lattice occupancy (d-HO sub-mode). We termed the coincidence between specific distributions of step sizes and the modes of motion as step-to-mode matching. Validation of this matching was obtained by randomizing the step directions of the experimental trajectories to break the matching of the diffusional modes to the original step sizes (the $S_iA_r$ simulated replicates). We found that this breaking caused the HE profiles to have partial rather than full overlap with that of the experimental replicates (Fig. 3g). Another validation was obtained by altering the step-sizes of the c-LO or the d-HO sub-modes at a time to break the matching of the step sizes to the original diffusive modes (the $c_{sh}$-LO and the $d_{lg}$-HO simulated trajectories). We found that manipulating the step-sizes caused well-defined deviations from the linear MSD-time profile (Fig. 9). Thus, we conclude that the combined behaviour of these non-random dynamics, rather than a simple stochastic process, is essential for the overall single-molecule behaviour to conform to the linear trend of MSD. This remarkable conservation of the linear trend suggests that these dynamics are in a subtle mechanistic balance, suggesting that they could be attributed to the same physical origin. While this physical origin is still not entirely clear, the good agreement between $\tau_{c-LO}$ and $\tau_R$ is noteworthy. This agreement partially accounts for the non-random dynamics and suggests that they are related to the conformational relaxation dynamics of DNA in which the relaxed conformations diffuse with shorter step sizes than do the compact conformations that diffuse with longer step sizes. Although a similar yet fundamentally different behaviour was previously elucidated as autocorrelated fluctuations in step sizes, these fluctuations are caused by changes in the radius of gyration and are attributed to the internal conformational fluctuations of DNA[11]. These internal fluctuations relax by diffusion, and their characteristic time is defined as the time required by the molecule to diffuse a distance that equals its radius of gyration $(t_{Rg}(\tau_{Rg}))$[29]. The relative dynamics that we report in this study can be distinguished from these internal fluctuations because the time scale of the relative dynamics is much longer than the characteristic time scale of the internal fluctuations $(\tau_{c-LO} \gg \tau_{Rg})$ (Fig. 2c, Supplementary Fig. 9b) and because of the characteristic step-to-mode matching that we observed in our analyses.

Further to the above-mentioned rational for the mechanism of the relative dynamics, we believe these dynamics could be related—in part—to the anisotropic diffusion of DNA because of its transient relaxation. During the time when the DNA molecule is relaxed, its shape is anisotropic, which can be modelled as an elongated ellipsoid where $a$ (length) $\gg$ $b$ (width). The resulting anisotropic diffusion involves two components, $D_a$, diffusion coefficient in directions parallel to the long axis $(D_a = k_B T / \gamma_a)$, and $D_b$, diffusion coefficient in directions perpendicular to the long axis $(D_b = k_B T / \gamma_b)$. Because the friction coefficient $\gamma_a$ is smaller than $\gamma_b$, $D_a$ is greater than $D_b$ and therefore the molecule is expected to show directed-like motion[30]. The time scale of this directed-like motion is determined by $\tau_\theta$ (the time required for the ellipsoid to diffuse 1 rad by rotational motion). At time scales longer than $\tau_\theta$, the rotation randomizes the motion and, eventually, results in a crossover from anisotropic diffusion to

isotropic diffusion[30]. Because the elongation of the DNA occurs transiently during the conformational relaxation process, we cannot gather conclusive evidence on whether $\tau_\theta$ is correlated with $\tau_R$ and with $\tau_{c-LO}$. Because of this uncertainty and the fact that shape isotropy/anisotropy cannot similarly explain the confined-like sub-mode motion, we argue that anisotropic diffusion could—only partially—explain the directed-like sub-mode motion. Taken together, we conclude that the conformational relaxation dynamics and the anisotropic diffusion partially elucidate the mechanism of the relative dynamics we describe here. Describing the full mechanism remains an open research question.

For the diffusion mode to be reliably captured using lattice occupancy analysis, the time scale of the dynamics should be slower than the frame rate of the detector and faster than the diffusion of the molecule out of the focal plane of the microscope. This limits the time scale of the dynamics that can be captured by lattice occupancy analysis. A detector with a faster frame rate[31], stroboscopic laser excitation[32] and 3D single-molecule tracking techniques[33] may further expand the applicability of the analysis to wider time scales.

The results reported here demonstrate that studying the relative motion of single molecules provides information on the dynamics hidden in their diffusive motion. These dynamics, which we term conserved linear dynamics, were not previously observed in the motion of single molecules. Our identification of conserved linear dynamics suggest that the apparent random diffusive motion of molecules in nature could actually be governed by non-random dynamics. Our observations and our analytical approach provide a new method for advancing our understanding of diffusion processes that are central to studies in diverse scientific fields. For example, from studying anomalous diffusion processes in biophysics to studying dynamic disorder in polymer science, our analytical approach could uncover essential dynamics and hence could provide access to intriguing applications. To that end, understanding the relative motion of molecules in terms of the specific modes of diffusion in the relevant fields (similar to what is shown in Figs 4,8 and 9) is essential. Such fundamental knowledge could also provide essential information on crucial diffusion-limited processes of the cell.

## Methods

**Materials.** Supercoiled ColE$_1$ (6.6 kbp) DNA was obtained from Nippon Gene (Toyama, Japan) whereas the lambda phage DNA was obtained from New England Biolabs (Hitchin, UK). The restriction enzyme SmaI and the digestion buffer were obtained from New England Biolabs and were used to prepare the linear form of the ColE$_1$ DNA. The DNA molecules were covalently labelled with Cy5 using a Label IT Cy 5 labelling kit obtained from Mirus Bio (Madison, WI, USA).

Suncoast yellow fluorescent polymer nanospheres (excitation/emission maxima 540/600 nm) of nominal size (0.19 μm; $2.653 \times 10^{12}$ nanospheres ml$^{-1}$) were purchased from Bangs Laboratories, Inc. (Fishers, IN, USA). The nanospheres were diluted with 70% glycerol in 10 mM TRIS buffer (pH 8) to yield a concentration of $1.5 \times 10^6$ nanospheres ml$^{-1}$.

**Preparation of the linear form of the supercoiled DNA.** To prepare the linear form of ColE$_1$, 8 μg of the supercoiled form were mixed with 50 units of SmaI in 50 μl of the digestion buffer (50 mM potassium acetate, 20 mM Tris-acetate, 10 mM magnesium acetate, 1 mM DTT, pH = 7.9). The reaction mixture was incubated at 25 °C for 8 h before removing the enzyme and the buffer components using isopropanol precipitation.

**Isopropanol precipitation.** The SmaI enzyme and the buffer components were removed after enzymatic digestion by the standard isopropanol precipitation method. Twenty-five microlitres of sodium acetate solution (3 M) and 40 μl of isopropanol were added to the digested DNA solution followed by ultra-centrifugation at 15,000 r.p.m. for 20 min at 4 °C. The supernatant was carefully removed and the DNA pellets were washed three times with 70 % ethanol. The washing step was repeated three times before the DNA pellets were dried in air.

**Labelling the DNA with Cy5 dye.** The $ColE_1$ DNA was covalently labelled with Cy5 dye according to the protocol accompanying the labelling kit. The DNA pellets obtained after isopropanol precipitation were dissolved in 37 μl water. Then, 5 μl of the labelling buffer and 8 μl of the labelling reagent were added to the DNA solution. The mixture was incubated at 37 °C for 2 h. The DNA was purified from the labelling reagents using isopropanol precipitation as described above. The labelled DNA pellets were dissolved in Tris EDTA (TE) buffer (10 mM Tris, 1 nM EDTA, pH = 8) to yield a concentration of 0.2 μg ml$^{-1}$. These labelling procedures should yield labelling efficiency of approximately one label every 10–30 base pairs according to the manufacturer's specifications.

**Preparation of the DNA imaging buffer.** To record single-molecule trajectories of appropriate lengths (more than 100 frames), the diffusion constant ($D$) was slowed from $D = 1.3$ μm$^2$ s$^{-1}$ (in TE buffer)[17] to $D = 0.17$ μm$^2$ s$^{-1}$ by adding glycerol to the imaging buffer. One hundred microlitres of glycerol was mixed with 78 μl of TE buffer and then degassed for 1 h. Then, 13 μl of an antioxidant cocktail (6 μl of 0.1 μM PCA, 6 μl of 1 μM PCD and 1 μl of 1 nM Trolox)[34] was added directly before the imaging experiment. Then, 5 μl of the 0.2 μg ml$^{-1}$ DNA solution was mixed with 5 μl of the imaging buffer to yield a final glycerol concentration of 25%. The solution was then sandwiched between a clean coverslip and a glass slide and sealed by a double-sided adhesive (0.12 mM, Grace-Biolabs, Bend, OR, USA). The labelled lambda DNA was dissolved in TE buffer at a concentration of 0.1 μg ml$^{-1}$. The calculated $D$ from the MSD plot was 0.29 μm$^2$ s$^{-1}$.

**Single-molecule fluorescence imaging measurements.** The single-molecule fluorescence imaging experiments were conducted on a custom-built epifluorescence microscopy setup[17]. The setup is based on an inverted microscope (IX71, Olympus, Tokyo, Japan) illuminated with a CW 100 mW 532-nm laser (Samba, Cobolt, Solna, Sweden) and a CW 60 mW 640-nm laser (MLD, Cobolt). The 532-nm and the 640-nm lasers passed through FF01-530/11 and LD01-640/8 excitation filters, respectively (Semrock, Lake Forest, IL, USA). The lasers were introduced into the microscope through two 5 × beam expanders (Thorlab, NJ, USA) and then through a focusing lens ($f = 300$ mm). The 532- and 640-nm laser lines were reflected to a UAPON 100XO TIRF NA 1.49 oil immersion objective lens by Di01-R532 and FF660-Di02 dichroic mirrors (Semrock), respectively. By means of an acousto-optical tunable filter (AOTF, AA Optoelectronics), the output of the excitation lasers was synchronized to an iXon Ultra EMCCD camera (Andor Technology, Belfast, Ireland) to illuminate the sample only during image acquisition and thus to minimize photobleaching. After illuminating the sample with either the 532- or 640-nm laser lines, the fluorescence from single molecules was collected by the same objective lens and then passed through FF01-580/60 and LP02-664RU emission filters (Semrock), respectively, before being introduced into the camera. All single molecule fluorescence images were recorded at a 0.16-μm pixel size and at 156 Hz with a 6.4 ms exposure time.

**Single-molecule localization and tracking.** Analysis of the single-molecule images of the diffusion trajectories was performed using a versatile tracking algorithm (see Supplementary Note 1 for details). This algorithm splits away the poorly localized data points from the trajectory and thus is applicable to studying the motion of macromolecules, such as the motion of chromosomes as well as the motion of small particles. A description of how we determined the position is in Supplementary Note 1. Ninety-eight single-molecule movies of 304 ± 214 frames each (mean ± s.d.) (min = 101 frames, max = 1,097 frames) were imported into the Matlab software and then the 2D spatial positions were exported as text files.

**Simulation of a single-molecule random walk.** The simulated trajectories were constructed using a routine written in Matlab starting at $(x,y) = (0,0)$. The random step sizes were generated using a distribution function ($R$) expected from the normal diffusion theory of a Brownian particle:[26]

$$R = \frac{2r}{\langle r^2(\Delta t) \rangle} \exp\left[\frac{-r^2}{\langle r^2(\Delta t) \rangle}\right], \tag{5}$$

where $r$ and $\Delta t$ denote the step size and the time lag, respectively. The step directions (angles between successive displacements) were generated based on random angles between 0 and 360°.

**Probability of square lattice occupancy.** We performed lattice occupancy analyses of the diffusion trajectories using a routine written in Matlab (Supplementary Data 1, Supplementary Note 2). The spatial positions obtained from the tracking algorithm were mapped onto a square lattice of side-length ($m$), which equalled the pixel size of the camera ($m = 0.16$ μm). The experimental probability was fitted to equation (4) to calculate the $P_{25}$ value.

**Detrended fluctuation analysis.** For the DFA to be statistically robust, we joined the single-molecule trajectories end-to-end. The combined trajectories of both the DNA and the nanospheres have an approximate length of $N = 30,000$ $\Delta t$. The

DFA[23] was performed using a routine written in Matlab (Supplementary Note 2). We generated integrated time series of $P_{25}$ ($P_{25}$ ($k$) ($k = 1,2,\ldots, N$)), $Z(i)$ by subtracting the mean $P_{25}$ value ($\langle P_{25} \rangle$)5 and integrating the time series:

$$Z(i) = \sum_{k=1}^{i} [P_{25}(k) - \langle P_{25} \rangle]. \tag{6}$$

The profile $Z(i)$ of length $N$ was then divided into non-overlapping segments ($s$) of equal size ($l$). The local trend in each segment ($Z_l(i)$) was calculated by subtracting the linear fit of the data:

$$Z_l(i) = Z(i) - f_s(i), \tag{7}$$

where $f_s(i)$ is the linear fit value in the $s$th segment. The root mean square variation (RMSV) for the segment size of $l$ (RMSV($l$)) was calculated by

$$\text{RMSV}(l) = \sqrt{\langle [Z_l(i)]^2 \rangle}. \tag{8}$$

We next calculated the power law exponent (the HE) that quantifies how fast the $RMSV$ grows as the segment width increases[23,35]. The HE were calculated by using logarithmically spaced segment sizes ($l = 50\Delta t - 5,000\Delta t$). The width of the shortest detrending segment ($50\Delta t = 0.32$ s) was set to be sufficiently longer than the time required by the DNA molecule to diffuse its radius of gyration ($\tau_{Rg} = 0.05$ s, Fig. 2c) so that the analysis would target the linear regime of Brownian motion (diffusive Brownian motion). The width of the longest detrending segment was set to 5,000$\Delta t$ (32 s) because it approximately represents 1/6 of the full length of the experimental replicate (30,000$\Delta t$) and hence the data from six segments can be averaged to give a statistically valid RMSV value.

HE of the $P_{25}$ temporal profiles recorded at different time lags ($i\Delta t$) ($i = 1,2,\ldots, 10$) (Fig. 3c) were obtained by calculating the RMSV at each $i\Delta t$ with segment sizes of $l = 50\Delta t - (5,000/i)\Delta t$. Cumulative HE (Fig. 3e) were obtained by subtracting the HE of a pure random walk (HE = 0.5) from the calculated HE values at each time lag followed by integrating those subtracted values. Note that, in the case of a random walk, such as the walk of nanospheres, the calculated HE is larger than 0.5 due to the local broadening of the probability time series, which is caused by the width of the sliding window.

**Mean-squared displacement analysis.** MSD was calculated by using the following expression:

$$\text{MSD}(\Delta t) = \langle (x_{i+n} - x_i)^2 + (y_{i+n} - y_i)^2 \rangle = 4D\Delta t, \tag{9}$$

where $x_{i+n}$ and $y_{i+n}$ describe the spatial positions after time interval $\Delta t$, given by the frame number, $n$, after starting at positions $x_i$ and $y_i$. $D$ is the diffusion constant. The theoretical MSDs at 1$\Delta t$ (MSD-1$\Delta t$) and at 10$\Delta t$ (MSD-10$\Delta t$) were calculated from the theoretical diffusion constant ($D$) by using equation (9). The theoretical $D$ was calculated by using the experimental MSD value at 1$\Delta t$ and 10$\Delta t$.

To generate a temporal profile of the absolute diffusion modes, we used the experimental replicates and the $S_rA_r$ replicates to calculate the MSD-$\Delta t$ profile of a sliding window of width 50$\Delta t$. The MSD-$\Delta t$ profiles (time lags between 1$\Delta t$ and 25$\Delta t$) were fitted to the normal (equation (9)), directed[36] (equation (10)), and confined[37] (equation (11)) diffusion models.

$$\text{MSD}(\Delta t) = 4D\Delta t + v^2(\Delta t)^2, \tag{10}$$

where $v$ is the drift velocity,

$$\text{MSD} = \frac{L^2}{3}\left[1 - \exp\left(\frac{-12D\Delta t}{L^2}\right)\right] + 4\sigma_{xy}^2, \tag{11}$$

where $L$ is the side length of the confined area and $\sigma_{xy}$ is the positional accuracy in $x$ and $y$ dimensions. The directed- and confined-like motions were quantified by $v$ and $L$, respectively. Diffusion modes whose MSD-time profile fitted to a linear trend or those that showed extreme irregularities that could not be fitted by using equations (9)–(11) were considered with no specific diffusion mode. We set a side-length limit of $L = 700$ nm (equation (11)) to avoid mistakenly classifying the MSD-$\Delta t$ plots. This limit, which approximately equalled $4R_g$, was empirically driven from the MSD-time profile and represented the maximum side-length that could describe confined-like motion of DNA within the 50$\Delta t$ window. When the fitting operation of the MSD-time profile to equation (S6) retrieved $L$ larger than 700 nm, we considered the chi-square of the linear (equation (9)) and the directed (equation (10)) motion profiles.

**Dynamic time warping analysis.** The DTW analysis was performed using a routine written in Matlab (Supplementary Note 2). The DTW algorithm[20] was used to generate a time-dependent motion profile illustrating the similarities and differences between the $P_{25}$ and the MSD time series within a sliding window of width 50$\Delta t$. The pairwise Euclidean distance ($d$) from each data point ($i = 1,2,3,\ldots$, $W$) of the normalized MSD time series ($Y$) to all the points ($i = 1,2,3,\ldots$, $W$) of the normalized $P_{25}$ time series ($X$) was calculated by

$$d(i,j) = |X_j - Y_i| \tag{12}$$

A matrix, $M$, of size ($W$, $W$) was constructed such that the value that the DTW algorithm recovered at position $M$ ($i_w, j_w$) of the matrix $M$ was the one with the

minimum cumulated distance:

$$M(i_w, j_w) = \min \sum_{w=1}^{W} d(i_w, j_w), \quad (13)$$

where $w$ is an integer between 1 and $W$ and it represents the corresponding position of $i$ and $j$ in matrix $M$. Dynamic programming was initialized such that the cumulated distance $M(i_w, j_w)$ was recursively calculated based on both the minimum value from the previous cumulated distances $[M(i_w - 1, j_w - 1), M(i_w - 1, j_w), M(i_w, j_w - 1)]$ and the distance recovered from the pair $d(i_w, j_w)$:

$$\left\{ \begin{array}{l} M(1,1) = d(1,1) \\ M(i_w, j_w) = [d(i_w, j_w) + \min\{M(i_{w-1}, j_{w-1}), M(i_{w-1}, j_w), M(i_w, j_{w-1})\}] \end{array} \right\}. \quad (14)$$

The output of the DTW algorithm of each window, $W$, was a single value accumulated at $M(W, W)$ suggesting how close both the MSD and the $P_{25}$ time series are. When the normalized profile of the MSD time series is higher in magnitude than that of the $P_{25}$ time series, the amplitude of local variation (ALV) is a positive value (equation (15)). Conversely, when the profile of the $P_{25}$ time series is higher in magnitude, the ALV value is a negative value (equation (15)). The ALV value approaches zero as the two motion profiles become similar:

$$ALV = \frac{\sum_{i=1}^{W} Y_i - \sum_{j=1}^{W} X_j}{\left| \sum_{i=1}^{W} Y_i - \sum_{j=1}^{W} X_j \right|} M(W, W). \quad (15)$$

**Conformational dynamics of DNA.** The conformational relaxation time of $ColE_1$ DNA ($\tau_R$) was estimated by the cumulative-area tracking method[17]. We analysed single-molecule movies with signal-to-background ratios greater than 3 for at least 500 consecutive frames. The single-molecule images were converted to binary images by using the tracking software. In the binary images, the pixels that are set to 1 identify the local maxima of the original images and hence they define the area occupied by the DNA molecule ($A_f$). The binary images were used to calculate the time dependent-fluctuations of $A_f$. The area fluctuations of approximately 35 movies (500 frames each) were autocorrelated and then the autocorrelation plots were averaged into a single autocorrelation plot. To account for the fluctuations that could arise due to brief partial escape of the molecule from the field of view (defocusing fluctuations), we analysed the polymer nanospheres using the same analytical method. The characteristic time of the defocusing fluctuations of the nanospheres ($\tau_n$) was estimated by fitting the averaged autocorrelation plot to a single-exponential decay using the following formula:

$$G(\tau) = B\exp(-t/\tau_n). \quad (16)$$

Because the depth of the field of view was the same for the DNA and the nanospheres, the characteristic time of the defocusing fluctuations of DNA ($\tau_d$) could be approximated by calculating the time required by the DNA to have an MSD value equal to that of the nanospheres using the equation

$$D^{DNA}/D^{Nsph} = \tau_n/\tau_d, \quad (17)$$

where $D^{DNA}$ and $D^{Nsph}$ are the diffusion constants of the DNA and the nanospheres, respectively. $\tau_R$ was estimated by fitting the averaged autocorrelation plot to a double-exponential decay using the following formula:

$$G(\tau) = B\exp(-t/\tau_d) + C\exp(-t/\tau_R). \quad (18)$$

**Data availability.** The data that support the findings of this study are available from the corresponding authors on request.

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

## Acknowledgements

The research reported in this publication was supported by funding from King Abdullah University of Science and Technology (KAUST) and the KAUST Office of Sponsored Research (OSR) under Award No. CRF-2015-2646-CRG4. We would like to thank Matthijs van Waveren, Antonio M. Arena and Alain Clo of KAUST IT Research Computing and Amine El Helou of MathWorks Ltd for their precious help in speeding up the MATLAB analysis and for providing the KAUST high performance computing (HPC) Add-on for the direct submission of the MATLAB script to the KAUST Noor

computer clusters. We thank Virginia Unkefer and Lina Mynar for editing the manuscript.

## Author contributions

M.F.S developed the theory, conceived and reached the scientific target, and devised the analytical approach. M.F.S. and S.H. wrote the manuscript. S.H. supervised the study.

## Additional information

**Competing interests:** The authors declare no competing financial interests.

