## [Peer Review File · Nature Communications]

Reviewers' comments:

Reviewer #1 (Remarks to the Author):

Review of "Conserved Linear Dynamics of Single Molecule Motion"

This manuscript focuses on the diffusive dynamics of macromolecules, in particular DNA. The authors show that their new method, based on measuring statistics of visiting new sites on a regular lattice, can yield more information than a mean-square-displacement measurement. The most important results it that with this method the authors show that, perhaps surprisingly, confined-like diffusion is associated with a low lattice occupancy and large step sizes, while directed-like diffusion is associated with a high lattice occupancy and small step sizes.

The topic is certainly interesting. However, I had great difficulty reading the manuscript.

First of all, some sentences are very unclear. An example can be found in the abstract: "We found that the speed of motion of DNA molecules has a non-intuitive effect on their mode of motion and uncovered that this effect backs up the mechanism of this conservation of the linear regime." For the uninitiated reader this is totally unintelligible.

Moreover, the text is often repetitive while at the same time concepts are not explained in a very precise way. For example, the precise (mathematical) definition of the probability P_i of visiting new sites is difficult to get from the paper. The supplementary material gives a definition, $P_i = \langle k_i \rangle / i$, but the exact meaning of $\langle k_i \rangle$ (as the average number of visits to new lattice sites) and i itself can only be understood after a very long and hard look at Supplementary figure 1.

About the method itself: its advantages or disadvantages compared to comparable techniques such as cumulative-area (CA) tracking should be explained. With the method described in this manuscript, I am not sure about the universality of the power-law behaviour and the exponents β obtained from the fits, e.g. in Supplementary figure 3. Does the probability always decay to zero, and can the authors prove that the β is independent of the range of frames that is included in the fit (in this figure 3 i runs from 3 to 25)? For example, for directed motion I would expect that ultimately, for high frame count, the number of newly visited sites would grow linearly with frame count, so P_i should become a constant. This actually seems to be the case in Fig. 3b for $i > 10$. Now a power-law fit is forced through this data. If the maximum frame number would have been larger, an exponent β closer to 0 would have been found.

In summary, the manuscript must be improved a lot before it is acceptable for Nature Communications.

If the authors are allowed to improve on all abovementioned points, I would appreciate an additional discussion on the possibility that conformational fluctuations of the DNA naturally lead to fluctuations in diffusion and an anisotropic diffusion tensor. Only averaged over larger time-scales the DNA may have a more or less isotropic shape, with isotropic diffusion. However, when looking instantaneously, the DNA is always non-spherical. At times it may even be elongated, with an associated anisotropic AND lower-than-average diffusion coefficient. This is consistent with the simultaneous observation of directed-like diffusion and smaller step sizes/higher occupancy, as described in this manuscript.

Reviewer #3 (Remarks to the Author):

In this article, the authors performed single-molecule tracking of large linear DNA. The authors found non-random dynamics, which could be detected in the relative motion, could govern the apparently random motion of DNA. This paper is interesting and the methods the authors provided would be useful for future research. However, in my opinion, this paper is not significant enough for its publication in nature communications in its current form.

My major concerns are:

1. The authors used 0.12 mm thickness channel. Thus, DNA has 3D diffusion. However, the authors could observed only 2D projection. Thus they used only 2D tracking for the data analysis. It should be addressed that 2D projection has no effect in their analysis for 3D diffusing DNA.
2. The data in Fig. 1a. is not a single-dye labeled DNA. It seems that the DNA labeled with multiple dyes (if not, the author should address why fig. 1a shows multiple spots). If DNA is labeled with multiple dyes, it is not easy to determine the location with DNA in high accuracy. How did the authors determine the positions of DNA in Fig. 1a with such blurring images?
3. The authors compared the diffusion of DNA with that of spherical polymer. To support the authors' claim and to improve the quality of this paper, the author should use DNAs that have different radius of gyration, such as the DNAs longer than or shorter than linear CoE1 DNA. Then, the comparison of these data will give more clear idea on the non-random dynamics.

Reviewer #4 (Remarks to the Author):

I have been invited at short notice to review the manuscript "Conserved Linear Dynamics of Single Molecule Motion" that has been submitted for publication in Nature Communications. Albeit I am somehow familiar with diffusion, single-particle tracking and single-molecule detection, the complexity and length of the manuscript (including a supplement of almost 50 pages) make an adequate and timely assessment rather difficult and I will therefore discuss only a few selected parts and issues. At this point, I am not convinced that the manuscript in its present form offers sufficient rigour and ingenuity to warrant publication in a high profile, multi-disciplinary journal like Nature Communications.

In their work, the authors use fluorescently labelled, linear DNA (6.6 kbp) and follow the molecular motion of the DNA using fluorescence imaging. The single-molecule data suggest that the apparently random motion of macromolecules such as DNA is in fact governed by non-random dynamics and the authors go to great lengths to disentangle and describe their findings.

Figure 1 and related text:

The author extensively analyse both mean square displacement (MSD) and the lattice occupancy, which describes the probability that the DNA molecule can be found in a new area after a given time. It is currently not clear, however, how the authors exactly determined the position of molecule to calculate the MSD. The authors mention the use of a "versatile tracking algorithm", but do not provide further details which are, in my opinion, essential in the context of the paper: The shape of the imaged DNA is clearly not circular, which would exclude the use of a 2D Gaussian to fit the position of the molecule as it conventionally done for point-like particles (either being single emitters or spherical beads). How was the 2D position of each DNA molecule obtained in figure 1? Does the blue circle

present the “centre of mass” of the molecule or the position of the simplified 2D Gaussian fit? In figure 1c, the author plot the theoretical MSD-Dt profile for the radius of gyration obtained from literature. Do the authors expect exactly the same radius of gyration under their experimental conditions featuring 25% of glycerol in the buffer as well as, potentially, modified dynamics and rigidity of the DNA due to the hundreds of fluorophores attached to the DNA (By the way, why were the spherical beads imaged in 70% glycerol as mentioned in the supplement and indicated by the 3x lower diffusion coefficient in sFig 5)? What values of the radius of gyration would a linear fit of the blue and red circles give? Maybe the authors could include the residuals of the linear fit to indicate more clearly the deviations (so they exist as the authors claim).

Figure 2 and related text:

I would suggest that the authors include a short definition and description of the Hurst exponent (HE) in the main text to guide the reader through the next section. In figure 2a, the authors compared the cumulative HE of the experimental data with the HE of various replicas in which either the step-directions, the step-distances or both were randomly ordered based on the original data. The authors show that only the data in which the angles were randomised did not lead to a loss in correlation. For Figure 2c I would have therefore intuitively expected that the randomly ordered step directions show the loss of MSD comparable to the experimental data, that initially showed the sub-diffusion (again, Figure 2c might benefit from linear fits and plotted residuals). Could the author clarify that finding? I would also suggest that the authors include comparisons with data on spherical beads to the supplement as an important control measurement.

Figure 3 and related text:

From what I understood, the authors took the experimental data (figure 1a: single, non-stitched time trace?) describing the mean position of the DNA as a function of time and fitted three different models of diffusion to a number of consecutive time points defined by a sliding window. The goodness of the fit then decides which model is most likely but no information is provided in whether the drift velocity or the positional accuracies have been fixed or not. Could the authors elaborate on their decisions? A free fit of the velocity, for example, could lead to situations where a low value of v makes a decision between a standard and directed diffusion rather arbitrary. On the other hand, fixing v to a certain value also seems arbitrary. Shouldn't be in any case the MSD (Dt) be calculated including the term for the localisation precision especially in hindsight which is here, as reasoned above, probably mainly coming from the non-Gaussian emission profile of the DNA? Similarly, the authors decided to limit the L parameter necessary for describing confined diffusion to 700 nm (roughly four times the radius of gyration). The question remains how robust the analysis is or whether by tuning individual parameters the result of the MSD analysis would correlate with other modes of relative motion. Furthermore, the authors classify four different sub-modes in figure 3 (c-LO, d-LO, d-HO, and c-HO), but continue to discuss mainly c-LO and d-HO. What happened to the other sub-modes? Are they here irrelevant?

Again, it would be interesting to see the comparisons between DNA and bead data diffusing under similar conditions.

Additional comments:

- It might be interesting at a later point to compare the results of the DNA labelled with Cy5 with data from DNA stained with an intercalating dye. One would expect that the observed time scales describing the modes of relative motion would shift especially if the density of intercalators is high.
- In the current analysis, any potential effects regarding 3D movement of the DNA, transient DNA-

surface interactions, photo-physical effects of the Cy5 emitters (blinking, photo-bleaching, self-quenching,...) have not been taken into account. This simplification is likely to be valid, but highlights the difficulty on interpreting the results especially concerning the dynamics.

Reviewers' Comments:

Reviewer #1:

Remarks to the Author:

The authors have significantly improved the presentation of their work, clarifying the abstract, and giving better explanation of the basic concepts of their approach. The advantages and disadvantages relative to CA tracking are now also highlighted.

However, I have a problem with their conclusion that friction anisotropy cannot explain the observed directed motion. The explanation hinges on their statement that the time scale of reorientation of the major direction of fraction anisotropy (τ_{θ}) is much smaller than the conformational relaxation time (τ_R) of the molecule. I am not convinced that this must be the case. For a random coil (or long wormlike chain) I think these two time scales should be of the same order of magnitude. Evidence of this can be found in the authors' own supplementary video of DNA motion: instantaneously, the distribution of the fluorescent dye is clearly non-spherical (at the beginning of the video it has a longest axis somewhat along the vertical direction). This is what one can expect for a random coil or long wormlike chain. The question is how fast does this longest axis decay to isotropy or reorient? The video shows that during the time the center-of-mass (blue point) diffuses over a distance equal to the typical radius of the fluorescent distribution (I assume this distance is already larger than or at least equal to the radius of gyration of the DNA molecule), the longest axis has not changed appreciably. In other words, I think the authors in their explanation unjustly neglect the correlation between the anisotropic mass gyration tensor and the anisotropic center-of-mass diffusion tensor.

Johan Padding

Reviewer #3:

Remarks to the Author:

In this revised article, the authors newly performed measurement on lambda DNA as a control and compared two different localization methods for analyzing their images. These new measurement and analysis addressed my concerns before successfully. Thus, I recommend its publication in Nature communication. However, the readability of this paper has not been improved yet.

Reviewer #4:

Remarks to the Author:

The revised manuscript "Conserved Linear Dynamics of Single Molecule (Brownian) Motion" has been considerably improved and my comments and suggestions have been carefully addressed. However, the manuscript is still very difficult to read and understand; the authors decision to move a significant part of the supplemental material into the main text did not help in that respect. My requested explanation of the Hurst coefficients, to give an example, introduced a new sentence which is difficult to understand: "Such fluctuations can be captured by detrending the temporal fluctuations by using detrended fluctuation analysis and by calculation of the Hurst exponent (HE)". I therefore strongly suggest another round focusing on readability and clarity.

Localisation/Tracking: The added explanation on the chosen algorithm is helpful, especially as the authors compared the centre of mass approach with the finally used LTA (abbreviation was not introduced!) algorithm and found similar results.

The authors further decided to include data from lambda phage DNA, but could not show any diffusion sub modes as the increased radius of gyration shifts the crossover point quadratically to longer times, which cannot be accessed experimentally. Now we have the situation that we have two negative "controls" (lambda phage DNA and spheres), but only one positive control. Maybe the authors can estimate which length scales under which conditions they expect to show sub modes?

REVIEWERS' COMMENTS:

Reviewer #1 (Remarks to the Author):

The authors have dealt with my last remaining issue, regarding the correlation between anisotropic mass distribution and anisotropic diffusion and their relevant timescales, in a satisfactory manner. I agree with their conclusion. I have no further comments.

Reviewer #4 (Remarks to the Author):

The authors have again improved their manuscript (especially the readability) and I therefore recommend the manuscript for publication in Nature Communications.

One more note, an effective time resolution considerably faster than the frame time of any camera can be achieved using stroboscopic laser excitation (<http://dx.doi.org/10.1039%2FC5CP04137F>).

Response to reviewers' comments:

We appreciate the thorough reading and detailed, constructive feedback from the reviewers that allowed us to more clearly explain and validate our findings. The numbering of the figures and the equations in this file uses the same numbering of the comments of the reviewers. All changes are shown in blue through the manuscript.

1) Response to reviewer #1

This manuscript focuses on the diffusive dynamics of macromolecules, in particular DNA. The authors show that their new method, based on measuring statistics of visiting new sites on a regular lattice, can yield more information than a mean-square-displacement measurement. The most important results it that with this method the authors show that, perhaps surprisingly, confined-like diffusion is associated with a low lattice occupancy and large step sizes, while directed-like diffusion is associated with a high lattice occupancy and small step sizes.

1.1) The topic is certainly interesting. However, I had great difficulty reading the manuscript.

We thank the reviewer for the constructive feedback. To enhance the readability of our manuscript, we divided the results into 6 titled-sections and rewrote the text. We also modified the figures for better readability.

1.2) First of all, some sentences are very unclear. An example can be found in the abstract: "We found that the speed of motion of DNA molecules has a non-intuitive effect on their mode of motion and uncovered that this effect backs up the mechanism of this conservation of the linear regime." For the uninitiated reader this is totally unintelligible.

We thank the reviewer for pointing out that the abstract is unclear to the uninitiated reader. Indeed, it is important to clarify the story and the main finding in a simple way as early as in the abstract. Thus, we rewrote the abstract so it is now suited to the uninitiated reader.

1.3) Moreover, the text is often repetitive while at the same time concepts are not explained in a very precise way. For example, the precise (mathematical) definition of the probability P_i of visiting new sites is difficult to get from the paper. The supplementary material gives a definition, $P_i = \langle k_i \rangle / i$, but the exact meaning of $\langle k_i \rangle$ (as the average number of visits to new lattice sites) and i itself can only be understood after a very long and hard look at Supplementary figure 1.

Moreover, the text is often repetitive while at the same time concepts are not explained in a very precise way. For example, the precise (mathematical) definition of the probability P_i of visiting new sites is difficult to get from the paper. The supplementary material gives a definition, $P_i = \langle k_i \rangle / i$, but the exact meaning of $\langle k_i \rangle$ (as the average number of visits to new lattice sites) and i itself can only be understood after a very long and hard look at Supplementary figure 1.

We understand the importance of enhancing the readability of the manuscript. We revised the manuscript and introduced the basic concept of the lattice occupancy analysis in the first section of the results. We also minimized the repetitions in both the main text and the supplementary materials.

We recognize the importance of precisely clarifying the mathematical definition of P_t . Thus, we supported the manuscript with a supplementary scheme. The scheme can be viewed as a separate supplementary file (tiff stack) where the reader can browse the calculation steps using the arrows. This scheme will help the reader to understand the meaning of the parameters in equations 1, 3 and 4.

1.4) About the method itself: its advantages or disadvantages compared to comparable techniques such as cumulative-area (CA) tracking should be explained.

Our lattice occupancy analysis and the cumulative area tracking analyze different aspects of molecular motion. However they complement each other in characterizing the molecular dynamics. In the following paragraph, we outline the major differences between both methods.

The cumulative area tracking (CA tracking) method analyzes the diffusion constant of single molecules by relating the mean cumulative area difference to the elapsed time. The major advantages of this method are: (1) it circumvents the localization step of the molecule while employing a simple tracking of a limited number of pixels (proxy pixels) that define the position of the molecule. The cumulative area difference of these pixels can be used to calculate the diffusion constant by using a simple mathematical formula. (2) By controlling the number of proxy pixels, the conformational dynamics can be simultaneously analyzed. This was done autocorrelating the time-dependent profile of the molecular area. A major limitation of the method in calculating the diffusion constant is that it requires both careful selection of the frame rate and trade-off between the expected displacement size and the pixel size. A major difference between our lattice occupancy algorithm and the CA tracking is that the later cannot be used to calculate the 2D lattice occupancy because of the random shape of the proxy pixels. The continuous change in the shape of these pixels during motion makes the interpretation of the lattice occupancy cumbersome. Whereas, the effect of the change in the shape of proxy pixels is averaged out during the calculation of the diffusion constant, a large error can be introduced in the time dependent lattice occupancy profile. This error is introduced due to the shape fluctuations of the proxy pixels.

In contrast to the CA tracking, our lattice occupancy analysis uses the conventional, sophisticated localization algorithms to localize the position of the molecule. Therefore, the time-dependent profile of the lattice occupancy is free from shape fluctuation-related errors of the proxy pixels and is representative to how the molecule occupies the 2D latticed space. Another major difference between both methods is that: to allow for accurate determination of the diffusion constant by the CA tracking method, the initial slope ($1\Delta t$) of the cumulative area difference-time plot is considered. This time lag should be much faster than the relaxation time of the polymer molecule, and hence, the diffusional and the relaxation dynamics are decoupled during the calculations. Although, the lattice occupancy method was not used to calculate the diffusion constant, we note that we considered slower time scales ($25\Delta t$) not fast time-scales ($1\Delta t$). This allowed us to

characterize the time scale of the mode of the motion and correlate it to the conformational relaxation time of the molecule.

We included this discussion in the revised version of the manuscript

1.5) With the method described in this manuscript, I am not sure about the universality of the power-law behaviour and the exponents β obtained from the fits, e.g. in Supplementary figure 3. Does the probability always decay to zero, and can the authors prove that the β is independent of the range of frames that is included in the fit (in this figure 3 i runs from 3 to 25)? For example, for directed motion I would expect that ultimately, for high frame count, the number of newly visited sites would grow linearly with frame count, so P_i should become a constant. This actually seems to be the case in Fig. 3b for $i > 10$. Now a power-law fit is forced through this data. If the maximum frame number would have been larger, an exponent β closer to 0 would have been found.

We understand that the reviewer raises concerns about the rational of the limited number of frames which we used in our analysis and about the validity of the scaling exponent (β). We respond to both points below.

We thank the reviewer for raising this important comment on the effect of the number of frames on the value of β . To elaborate on this comment, we calculated the β values of 100 simulated random trajectories (displacement = 160 nm, pixel (lattice) size = 160 nm) at different number of frames ($n=50-1000$). As shown in figure 1.5a, the number of frames has a negligible effect on the value of β . However, we note that the β value is not included in our calculations and that a relatively small number of frames should be analyzed as explained in the following sections.

Fig. 1.5a. Effect of trajectory length on the value of the scaling exponent (β). The error bars are the standard deviations of 100 simulated trajectories.

Our findings are based on the time-dependent characterization of the motion of single molecules. Therefore, our calculations are based on analyzing the motion at discrete time intervals. To obtain good statistics, we chose to analyze 50 frames and, then, calculate the time-dependent P_{25} profile. To maintain enough temporal resolution of the motion, we cannot analyze more frames or even extrapolate the data to larger number of frames. Also, importantly, at larger number of frames the definition of the confined-like or the directed-like motion is no longer valid because the mode of motion tends to show linear MSD profile. In this case, the absolute and the relative modes which we observed in figure 4 are, indeed, averaged out at larger number of frames.

Regarding the scaling exponent β , we disregard the value of β because it is related to the absolute modes of the motion (confined-like and directed-like motion). Therefore, we probe the relative motion as a single P value (after fitting to equation 4) regardless of the modes of the absolute motion (i.e. regardless the value of β). Even if the fitting is forced for directed motion to give β value close to zero (although this is not the case of directed-like motion within normal diffusion because the zero indicates regular flow), we disregard this value and only consider the P_{25} value. If P_t fits to a linear equation (at $i > 10$), we would obtain P_{25} value equals to the value obtained from the fitting to equation 4 (figure 1.5b).

Fig. 1.5b. Analysis of the directed motion by using lattice occupancy analysis as shown in figure S2b. The green line is the linear fitting of the probability values at $i > 10$ to the linear equation ($y = a + bx$). The P_{25} values after the linear fitting and after the fitting to equation 4 are 0.6927 (≈ 0.69) and 0.6868 (≈ 0.69) respectively.

1.6) In summary, the manuscript must be improved a lot before it is acceptable for Nature Communications.

We hope that the manuscript in its present form satisfies the criteria of quality that merits the publication in Nature Communications.

1.7) If the authors are allowed to improve on all abovementioned points, I would appreciate an additional discussion on the possibility that conformational fluctuations of the DNA naturally lead to fluctuations in diffusion and an anisotropic diffusion tensor. Only averaged over larger time scales the DNA may have a more or less isotropic shape, with isotropic diffusion. However, when looking instantaneously, the DNA is always non-spherical. At times it may even be elongated, with an associated anisotropic AND lower-than-average diffusion coefficient. This is consistent with the simultaneous observation of directed-like diffusion and smaller step sizes/higher occupancy, as described in this manuscript.

We do agree with the reviewer that the conformational fluctuations could play a great role when the DNA molecule diffuses. These fluctuations, however, are not simple as continuous fluctuations due to elongation (relaxation) and shortening (compaction) of the shape of the molecule. As discussed in reference 14, slow conformational rearrangement of the molecule could also be at work. Although, we showed that the time scale of the non-random relative modes matches the time scale of conformational fluctuations, we believe that the c-LO and the d-HO sub-modes are not solely related to these simple fluctuations. We drive the evidence on this claim through the following two-point discussion.

- Discussion on anisotropic (directed-like) motion and its time scale of elongated molecules:

At the time where the molecule is relaxed (elongated), we do agree with the reviewer that the diffusion is anisotropic because the DNA is non-spherical. In such a case, the shape of DNA can be modeled as elongated ellipsoid (where a (length) $\gg b$ (width)). Thus, the anisotropic diffusion of this elongated structure involves two components (D_a : diffusion in directions parallel to the long axis ($D_a = k_B T / \gamma_a$) and D_b : diffusion in directions perpendicular to the long axis ($D_b = k_B T / \gamma_b$)). Because the friction coefficient γ_a is less than γ_b (Happel, J. *et al.* Low Reynolds Number Hydrodynamics (Kluwer, Dordrecht, Netherlands, 1991), D_a is greater than D_b

and therefore the molecule is expected to show directed-like motion (Han, Y. *et al.* Science 314,626-340 (2006)). The time scale of this friction anisotropy is determined by τ_θ (the time required for the ellipsoid to diffuse 1 rad by rotational motion). At time scales less than τ_θ , the directional motion is enabled (because D_a is greater than D_b). However at time scales longer than τ_θ , the rotation randomizes the motion, washes out the directional memory and, eventually, results in a crossover from anisotropic diffusion to isotropic diffusion (Han, Y. *et al.* Science 314,626-340 (2006)).

- Discussion on the time scale of the friction anisotropy of DNA compared with the conformational relaxation time: Because the elongation DNA is only transient during the whole conformational relaxation process (relaxed form \rightleftharpoons compact form), the time scale during which the molecule stays in elongated shape should be faster than the time scale of the whole conformational relaxation process, *i.e.* the τ_θ is much faster than the conformational relaxation time of the molecule ($\tau_\theta \ll \tau_R$). Because we found that the time scale of the relative modes approximately equals to the conformational relaxation time ($\tau_{\text{CLO}} \approx \tau_R$), we conclude that the directed-like mode, which we detected, is not solely related to the anisotropic diffusion at time instances where the molecule is elongated.

We added the above discussion in the revised manuscript.

Reviewer #3 (Remarks to the Author):

In this article, the authors performed single-molecule tracking of large linear DNA. The authors found non-random dynamics, which could be detected in the relative motion, could govern the apparently random motion of DNA. This paper is interesting and the methods the authors provided would be useful for future research. However, in my opinion, this paper is not significant enough for its publication in nature communications in its current form.

We thank the reviewer for the constructive and insightful feedback. The comments helped us greatly to improve the discussion and properly validate the findings.

My major concerns are:

3.1) The authors used 0.12 mm thickness channel. Thus, DNA has 3D diffusion. However, the authors could observe only 2D projection. Thus they used only 2D tracking for the data analysis. It should be addressed that 2D projection has no effect in their analysis for 3D diffusing DNA.

We believe that the nanoparticles —whose diameter (0.19 μm) equals to the radius of gyration of the DNA— addresses the 3D motion. Basically, we have not observed any relative dynamics (dynamics resulting from relative measurements) in the 3D diffusing nanoparticles. However, we agree with the reviewer and believe that the 3D motion should be also addressed for another 3D diffusing DNA. We show these results in our response to comment 3.3 below.

3.2) The data in Fig. 1a. is not a single-dye labeled DNA. It seems that the DNA labeled with multiple dyes (if not, the author should address why fig. 1a shows multiple spots). If DNA is labeled with multiple dyes, it is not easy to determine the location with DNA in high accuracy. How did the authors determine the positions of DNA in Fig. 1a with such blurring images?

The 6.6kbp DNA molecules were covalently labeled with Cy5 dye with labeling density of one label every 10-30 base pairs. Therefore, each single molecule should have a mean of 300 Cy5 labels. We used published single molecule localization and tracking algorithm (supplementary reference 28) to localize and track the single molecules. We provide further details on the localization and tracking algorithm in our response to comment 4.1 (below). Meanwhile, to address the concern of the reviewer about the impact of the accuracy of the localization on our findings, we compare the results based on the center of mass algorithm (COMA, calculation of the centroid position) with the results based on the localization and tracking algorithm (LTA).

To highlight the similarities/difference between the COMA and the LTA in our analyses, we consider the 6.6kbp molecule shown in supplementary movie 1. The distributions of the calculated displacements by the LTA and the COMA are shown in figures 3.2a-c. For clarity and to ease comparison, we show only the first 300 tracks of both calculations in figures 3.2d,e. We

found that the distribution of displacements obtained the LTA resulted in better fitting to the normal 2D diffusion theory (Eq. 5) compared with those obtained by the COMA (Figure 3.2c). Although the calculated displacement value (r), from the fitting, was found approximately the same in both calculations (Fig. 3.2a,b), we decided to track the molecules using the algorithm that resulted in a better fitting of the displacement data to the normal 2D diffusion theory (Eq. 5, figure 3.2c). This explains why we used the LTA and not the COMA in our analyses.

Indeed, we checked whether we could detect the same deviations in the relative motion in the tracks obtained using the COMA. To that end, we generated 100 experimental replicas and 100 S_rA_r simulated replicas from the COMA tracks. We applied the same analytical procedures as explained in the manuscript to detect and analyze the tracks for the relative motion of the DNA molecules. We found that the COMA resulted in cumulative HE profiles (Fig.3.2f) similar to those shown in the manuscript figure 3e (obtained from the TLA tracks). Therefore, we conclude that the minor differences between the LTA and the COMA tracks do not affect our results.

We added the above discussion in the supplementary materials of the revised manuscript.

Figure 3.2. Comparison of the calculations based on the LTA and the COMA. a,b) the distribution of displacements obtained by using the LTA (a) and the COMA (b). The calculated displacement value (r), after fitting the data (the red line) to equation 5, is shown for each algorithm. c) The residuals of fit of the displacement data in a,b to equation 5. The blue plot shows the residuals for the LTA, whereas the red plot shows the residuals for the COMA. d,e) Molecular tracks of the 6.6kbp DNA obtained by using the LTA (d) and the COMA (e). f) Cumulative Hurst exponents (HE) at different time lags ($i\Delta t$) of the relative modes of the experimental replicas (blue) and the S_rA_r (red) simulated replicas. These data were obtained from the COMA tracks. The data of the LTA tracks are shown in manuscript figure 3.

3.3) The authors compared the diffusion of DNA with that of spherical polymer. To support the authors' claim and to improve the quality of this paper, the author should use DNAs that have different radius of gyration, such as the DNAs longer than or shorter than linear CoE1 DNA. Then, the comparison of these data will give more clear idea on the non-random dynamics.

We thank the reviewer for pointing out this important issue. A comparison with DNA of larger radius of gyration can give more clear idea on the non-random dynamics. We compared lambda DNA whose radius of gyration is $0.7 \mu\text{m}$ (Smith, D. *et al* Phys Rev Lett 75,22,4146-4149) with our 6.6kbp DNA .

After analyzing the single molecule images of lambda DNA, we found that its diffusion coefficient is $D = 0.29 \mu\text{m}^2/\text{s}$. The MSD of Lambda DNA showed subdiffusion profile characteristic for the crossover regime of DNA (Figure 3.3a-b). We did the same analytical procedure on the lambda DNA tracks and then calculated the Hurst exponent of the relative motion of 100 experimental replicas. Figure 3.3c-f shows that the calculated Hurst exponent of the experimental lambda DNA replica and their S_rA_r replica at $1\Delta t$ (the time scale at which we detected the CLO and the DHO sub-modes). These results are also shown in a comparison with the calculated Hurst exponent of the 6.6kbp DNA. Because the Fourier transform analysis was successful —only— for the CLO sub-mode, we show —in figures 3.3g-h— the mode of motion of the CLO sub-trajectories of the lambda DNA tracks and its frequency-magnitude spectra. Our results show that: 1) there is no difference in the experimental Hurst exponents of Lambda DNA

compared with the simulated ones. This indicated no anomalous relative motion. 2) Neither the confined-like mode nor the strong frequency signal which we observed in the case of the 6kbp DNA was recovered by the analyses in the case of Lambda DNA. This further confirmed that the motion does not show the characteristic relative sub-modes which we observed during the motion of the 6.6kbp DNA molecules.

To interpret these observations, we note that: because the radius of gyration of lambda DNA is 3.7 times larger than that of the 6.6kbp DNA, the length scale we are analyzing —as shown in figure 3.3a-g— is shorter than the radius of gyration of lambda DNA. Indeed, we could not record very long trajectories to exceed the length scale of the radius of Gyration due to frequent conformational changes and photobleaching. Therefore, we conclude that the motion of lambda DNA within its radius of gyration does not show the type of relative dynamics which we are reporting in our study. We believe that this experiment and analyses —suggested by the reviewer— are perfect controls to our 6.6kbp results in terms of both the length scale of the dynamics and the 3D motion of DNA. Because the diffusion coefficient of 6.6kbDNA and that of lambda DNA are close to each other, we conclude that the relative dynamics which we observed are not artifacts due to the 3D motion.

We have included this important control in the revised manuscript.

Figure 3.3a-b. Single molecule tracking and analyses of the molecular tracks of lambda DNA by using mean squared displacement (MSD). a) A molecular track of lambda DNA molecules. The pixel size is $0.16 \mu\text{m}$. b) MSD- Δt profile of lambda DNA (blue circles, $\Delta t = 0.0064 \text{ s}$). The red line is the theoretical MSD- Δt profile.

Figure 3.3c-f. Frequency histograms of the Hurst exponent (HE)- $1\Delta t$ distribution of DNA experimental and their S_rA_r simulated replicas. The type of DNA and the mean values are indicated in each case.

Figure 3.3g-h. Characterization of the diffusional c-LO sub-mode of lambda DNA. g) The averaged MSD- Δt profiles of the sub-trajectories captured in the $-ALV$ zones of the first mode-set of the lambda DNA experimental replica (red circles; data that correspond to the c-LO submode). The black solid line shows the theoretical diffusion. The mode of these sub-trajectories agrees with a pure random behavior. The blue circles show the overall MSD profile (as shown in figure 3.3b) and the green line shows its theoretical diffusion. h) The frequency-magnitude spectrum of

the c-LO extended data set of lambda DNA experimental replica. The spectrum shows no periodic displacement signal in the c-LO extended data set.

Reviewer #4 (Remarks to the Author):

I have been invited at short notice to review the manuscript "Conserved Linear Dynamics of Single Molecule Motion" that has been submitted for publication in Nature Communications. Albeit I am somehow familiar with diffusion, single-particle tracking and single-molecule detection, the complexity and length of the manuscript (including a supplement of almost 50 pages) make an adequate and timely assessment rather difficult and I will therefore discuss only a few selected parts and issues. At this point, I am not convinced that the manuscript in its present form offers sufficient rigour and ingenuity to warrant publication in a high profile, multi-disciplinary journal like Nature Communications.

In their work, the authors use fluorescently labelled, linear DNA (6.6 kbp) and follow the molecular motion of the DNA using fluorescence imaging. The single-molecule data suggest that the apparently random motion of macromolecules such as DNA is in fact governed by non-random dynamics and the authors go to great lengths to disentangle and describe their findings.

We thank the reviewer for the thorough reading and for the insightful, in-depth comments on the manuscript despite the limited time. Based on the comment of the reviewer on the lengthy supplementary information, we shortened the supplementary information by moving most of these information to the proper places in the main text.

Figure 1 and related text:

4.1) The author extensively analyse both mean square displacement (MSD) and the lattice occupancy, which describes the probability that the DNA molecule can be found in a new area after a given time. It is currently not clear, however, how the authors exactly determined the position of molecule to calculate the MSD. The authors mention the use of a "versatile tracking algorithm", but do not provide further details which are, in my opinion, essential in the context of the paper: The shape of the imaged DNA is clearly not circular, which would exclude the use of a 2D Gaussian to fit the position of the molecule as it conventionally done for point-like particles (either being single emitters or spherical beads).

How was the 2D position of each DNA molecule obtained in figure 1? Does the blue circle present the “centre of mass” of the molecule or the position of the simplified 2D Gaussian fit?

We used published single molecule localization and tracking algorithm (supplementary reference 4: Jaqaman, K. et al. Nat Methods 5, 695-702 (2008)). The algorithm uses mixture-model fitting algorithm to localize and track multiple particles in the same field of view (Figure 3 in supplementary reference 4). Furthermore, it can detect particles’ merging and splitting during motion (Figure 4 in supplementary reference 4). To achieve these tracking targets, the algorithm localizes and tracks all the local maxima in the single molecule image including maxima that are partially overlapping. As discussed in reference 4, the algorithm is universal and can track long molecules as well as point-like particles. We provide the below details on how we exploited this algorithm to determine the position of the 6.6kbp molecule.

To illustrate the performance of this algorithm in our data analysis, we consider the molecule shown in supplementary movie 1. The single molecule image of this relatively short dsDNA contains up to 1-2 fluorescence maxima (Figures 4.1a,b). The algorithm detects and tracks both of them independently (Figures 4.1c). Figure 4.1a shows the DNA molecule where two fluorescence maxima were identified. These maxima (figure 4.1b) were independently fitted to 2D Gaussian function to determine the position (dark blue and green circles in figure 4.1a, more technical details on the robust fitting using the Gaussian kernel and the tracking algorithms can be found in figure 1, equations 1-6 and in supplementary notes 3 and 4 of supplementary reference 4). These two maxima were independently tracked as shown in figure 4.1c. If one of the fluorescence maxima is lost due to the escape from the field of view (Figure 4.1d-g), its tracking is terminated while the tracking of the other maximum is continued (Figure 4.1h). It is important to underline that: (1) in our analysis, we consider only one track per one molecule. We chose the track that corresponded to the brightest local maxima in the single molecule image (i.e. track 1 in figure 4.1h), (2) the reason for which we used the localization and tracking algorithm not the center of mass calculations is explained in comment 3.2.

We included the above discussion in the supplementary materials of the revised manuscript.

Figure 4.1a-b. Intensity profiles of the DNA shown in supplementary movie 1. The localization and tracking algorithm detected two fluorescence maxima. The dark blue and green arrows in b indicate fluorescence maxima 1 and 2 respectively. The dark blue and green circles indicate the position after independent fitting of both maxima to the Gaussian Kernel. The red arrow/point indicates the calculated center of the mass.

Figure 4.1c. The molecular tracks (frame 2 ~ frame 92) of the DNA shown in supplementary movie 1. The track number is shown for each image. The blue track is a track of fluorescence maxima 1 (see figure 4.1a,b), whereas the green track is a track of local maxima 2.

Figure 4.1d-g. Intensity profiles of frame 786 (d,e) and frame 787 (f,g) of the DNA shown in supplementary figure 1. The dark blue and green arrows indicate fluorescence maxima 1 and 2 respectively. The dark blue and green circles indicate the position after fitting using the Gaussian Kernel of the fluorescence maxima 1 and 2 respectively. The localization and the tracking algorithms detected two maxima in frame 786 but not in frame 787. In frame 787 only one maximum (1) was detected with a breakdown to the other maximum (2).

Figure 4.1h. The molecular tracks (frame 386 ~ frame 1086) of the DNA shown in supplementary movie 1. The track number is shown for each image. At frame 786, the tracking of trajectory 2 (green; fluorescence maximum 2) was terminated (see figure 4.1d-g), whereas the tracking of trajectory 1 (blue; fluorescence maximum 1) is continued.

4.2) In figure 1c, the author plot the theoretical MSD-Dt profile for the radius of gyration obtained from literature. Do the authors expect exactly the same radius of gyration under their experimental conditions featuring 25% of glycerol in the buffer as well as, potentially, modified dynamics and rigidity of the DNA due to the hundreds of fluorophores attached to the DNA. What values of the radius of gyration would a linear fit of the blue and red circles give? Maybe the authors could include the residuals of the linear fit to indicate more clearly the deviations (so they exist as the authors claim).

Here, we are responding to comments 4.2 and 4.4 together.

For DNA, the radius of gyration is calculated based on the worm-like chain model (Brinkers, et al. J. Chem. Phys. 130,215105 (2009)):

$$\langle R^2 \rangle = 2Pl \left[1 - \frac{P}{l} \left(1 - e^{-\frac{l}{P}} \right) \right] \xrightarrow[P=50nm, l=2178nm]{l \gg P} \langle R^2 \rangle = 2Pl \quad \text{Equation 4.2a}$$

$$\langle R^2 \rangle = 6 \langle R_g^2 \rangle \quad \text{Equation 4.2b}$$

R^2 is the end-to-end distance, P is the persistence length of DNA, l is the contour length of DNA and R_g is the radius of gyration. Thus, the end-to-end distance for the 6.6kbp DNA is $0.47\mu\text{m}$ and the calculated radius of gyration is $0.19\mu\text{m}$ (experimental $R_g = 0.186$ (Voordouw, G., et al.

Biophys Chem 8, 171-189 (1978))). The radius of gyration is therefore dependent on both the persistence length and the length of DNA. We note that, the time scale of the c-LO sub-mode is 0.33s which corresponds to a MSD value of $0.196\mu\text{m}^2$. Thus, the length scale of the c-LO sub-mode is 0.44 μm . This length scale corresponds to the radius of gyration of a 35kbp molecule (more than 5 times longer than our 6.6kb DNA molecule). Therefore and before going to the below discussion we highlight that the dynamics we are reporting occur at much longer length scale compared with the radius of gyration of the 6.6kb DNA molecule.

We understand that the reviewer wants us to calculate the radius of gyration from the MSD plot shown in the manuscript figure 2c (the red and blue circles) and to show the residuals of the fitting to the linear equation. In figure 4.2a,b, we show the residuals of the linear fitting to the MSD plot at $t > 8\Delta t$. The plot showed excellent fit to the linear equation and highlighted the deviation at time scales $< 8\Delta t$ (subdiffusion). To reliably calculate the radius of gyration from the fitting of the MSD plot in figure 4.2a, the MSD should be calculated in the hydrodynamic regime (Rouse/Zimm regime). To achieve this, the time resolution of the measurement has to be in the μs scale which is not accessible in widefield microscopy (accessible, for example, in a DLS measurement). We attempted, however, to calculate the radius of gyration from the MSD prediction model of Rouse and Zimm dynamics. This can be done by fitting the MSD to the following expression:

$$\langle r^2(t) \rangle = 2(6\pi DR_H t)^a \quad (\text{Shusterman, R. Phys. Rev. Lett 92,4, 048303 (2004)}) \quad \text{Equation 4.2c}$$

where D is the diffusion coefficient ($D=0.17\mu\text{m}^2/\text{s}$), R_H is the hydrodynamic radius and a is the scaling exponent ($a=1/2$ for Rouse dynamics and $a=2/3$ for Zimm dynamics). For semiflexible linear polymers, the radius of gyration can be calculated from the hydrodynamic radius by using the following expression:

$$\frac{R_{g(\text{Linear})}}{R_{H(\text{Linear})}} = 8 / 3\pi^{1/2} = 1.508 \quad (\text{Robertson, R. PNAS 103(19),7310-7314 (2006)}) \quad \text{Equation 4.2d}$$

Where R_g is the radius of gyration of the linear polymer.

Figure 4.2c,d shows the free fitting of the MSD profile of the 6.6kb DNA to equation 4.2c. The calculated hydrodynamic radius after the fitting is 0.076 μm and the calculated radius of gyration is 0.12 μm . The value is indeed less than the theoretical value 0.19 μm (equations 4.2a,b). This discrepancy is also evident in the calculated scaling exponent ($a=0.926$, figure 4.2c) which clearly indicates departure from the hydrodynamic regimes to the cross over regime.

Whereas, the viscosity does not affect the radius of gyration, the attachment of an intercalating dye increases the radius of gyration by 5% (Liu, Y. *et al.* *Macromolecules* 40,2172-2176(2007)). Although the contour length is increased 30-40%, the radius of gyration, after staining, is not increased that much. On the other hand the viscosity of the medium does affect the speed of molecular dynamics where the longest relaxation time (λ) is directly proportional to the viscosity of the medium through the following expression:

$$\lambda = \frac{0.422\eta_s[\eta]M}{RT} \quad (\text{Liu, Y. et al. Macromolecules 40,2172-2176(2007)}) \quad \text{Equation 4.2d}$$

where η_s is the viscosity of the solvent, $[\eta]$ is the intrinsic viscosity of the polymer, M is the molar mass, R is the gas constant and T is the temperature.

In our experiment we used covalently attached Cy5 (not intercalating) dye. This label is attached via a di-amine linker and then via a small reactive amino group to DNA. We modeled this structure as shown in figure 4.2e and then calculated the distance between the cy5 label and the DNA. We found that the length of the linker approximately equals to the width of the DNA itself. According to the specifications of the manufacturer, the attachment of the Cy5 label using this long linker does not affect the structure of DNA (<https://www.mirusbio.com/products/labeling/label-it-nucleic-acid-labeling-reagents>).

Therefore, taken together, we found that it is reasonable to refer to the reported value of the radius of gyration ($R_g=0.186\mu\text{m}$) at this viscosity and labeling conditions.

Figure 4.2a-d. Fitting of the MSD profile of 6.6 kbp DNA to the linear equation and to the hydrodynamic MSD prediction model. a) Fitting of the MSD profile shown in the manuscript figure 2c to the linear equation. The fitting was restricted to the linear part of the profile ($t > 8\Delta t$). The fitting was extended to $1\Delta t$ to show the deviation at the fast time scales ($1\Delta t - 8\Delta t$). The green line is the theoretical MSD. b) The fitting residuals of a. c) Fitting of the MSD profile shown in the manuscript figure 2c to the hydrodynamic prediction model (equation 4.2c). d) The fitting residuals of c.

Figure 4.2e. Modeling of the Cy5-linker which was attached to the 6.6 kbp DNA molecule

4.3) By the way, why were the spherical beads imaged in 70% glycerol as mentioned in the supplement and indicated by the 3x lower diffusion coefficient in sFig 5?)?

We imaged the spherical beads in higher glycerol concentration to slowdown the diffusion, so that the displacement value (36nm) of the nanoparticles is comparable with the mean localization error. Figure 4.3a shows the calculated localization accuracy (Thompson, R. *et al.* Biophys. J. 82,2775-2783 (2002)) after the 2D Gaussian fitting. The figure shows that the displacement value is 7 times larger than the maximum localization accuracy. We wanted to know whether the localization error could affect our analysis and could result in false positive relative dynamics. After carrying out the imaging at this condition, we found that the single molecular track is free of any anomalous relative motion (manuscript figure 3d (shown below)). Manuscript figure 3d shows the Hurst exponent (HE)- $i\Delta t$ profile of the nanoparticles in comparison with the random simulation. The plots show full overlap of the experiment and the simulation at all time scales. Importantly, the overlap is also demonstrated at $3\Delta t$ (62nm) which approximately corresponds to the displacement value of the 6.6kb DNA molecule. Therefore, at this length scale we did not detect any anomalous relative motion. Because the motion of nanoparticles is definitely random, we did this control experiment, in principle, to prove that the random motion of nanoparticles does not result in any false positive results due to localization errors. Figure 4.3b shows the localization accuracy of 6.6 Kbp DNA for a comparison.

Figure 4.3a,b. Localization accuracy of the nanoparticles and the DNA molecules. The black arrow indicates the displacement value of the nanoparticles whereas the red arrow indicates the displacement value of the 6.6kbp DNA.

Manuscript figure 3d (shown for convenience). Caption: Hurst exponent (HE)- $i\Delta t$ profiles of nanospheres (purple) and their S_rA_r replicas (light green). The HE of the experimental profiles were similar to those of the simulated profiles, and this indicates random fluctuations of the P_{25} values.

4.4) What values of the radius of gyration would a linear fit of the blue and red circles give? Maybe the authors could include the residuals of the linear fit to indicate more clearly the deviations (so they exist as the authors claim).

We combined this comment with comment 4.2. We responded to both comment together as shown underneath comment 4.2.

Figure 2 and related text:

4.5) I would suggest that the authors include a short definition and description of the Hurst exponent (HE) in the main text to guide the reader through the next section.

As recommended by the reviewer, we added the definition of the Hurst exponent in the revised manuscript.

4.6) In figure 2a, the authors compared the cumulative HE of the experimental data with the HE of various replicas in which either the step-directions, the step-distances or both were randomly ordered based on the original data. The authors show that only the data in which the angles were randomised did not lead to a loss in correlation. For Figure 2c I would have therefore intuitively expected that the randomly ordered step directions show the loss of MSD comparable to the experimental data, that initially showed the sub-diffusion (again, Figure 2c might benefit from linear fits and plotted residuals). Could the author clarify that finding? I would also suggest that the authors include comparisons with data on spherical beads to the supplement as an important control measurement.

We understand that the reviewer raises concerns about: **Case 1)** why in the case of random angles & same order of displacements, the MSD profile is linear (loss of subdiffusion). Whereas in the case of same order of angles & random displacements, the MSD is subdiffusive (subdiffusion is maintained). **Case 2)** why in the case of random angles & same order of displacements, the Hurst-exponent cumulative profile overlaps with the experimental data (Hurst correlation is maintained). However, in the case of same order of angles & random displacements, the Hurst-exponent cumulative profile overlaps with the fully random simulated data (loss of Hurst correlation). We explain both cases below.

Case 1:

In the manuscript figure 2c, randomizing step directions (red circles), indeed, resulted in trajectories that lacked the subdiffusion motional mode. Whereas, randomizing step lengths while keeping the step directions intact resulted in trajectories with subdiffusion motional mode (black circle). To clarify this point, we will consider the effect of step directions on the confined mode of motion in the below discussion. The below discussion can be extrapolated to discuss the subdiffusion and the directed mode of motion as well.

Typically, if a diffusing molecule translating inside a closed area of space is analyzed, the analysis would show that the step-directions are favored toward the interior of the closed area. Therefore, the order of step directions is closely related to the mode of the motion. We showed in the manuscript figure 3g that keeping the order of the angles intact caused the cumulative *HE* provide to be nearly indistinguishable from the fully random simulation (randomized step lengths and step directions). On the other hand, we showed in the manuscript figure 3h that only by keeping the order of the angles intact, the MSD plot retains its original mode (subdiffusion). Therefore, it is safe to conclude that the mode of motion is correlated with the order of the step directions but not with the order of step lengths (red circles). We further verify this in the below discussion.

Figure 4.6a-f show simulated data on the effect of randomizing the step directions and step lengths at a time on the MSD plot. These simulated results support our above conclusion on the

correlation of the mode of the motion with the order of the step directions but not with the

order of the displacements.

Manuscript figure 3g h (shown for convenience). g) Averaged values of the cumulative HE of the experimental and simulated replicas. The experimental replica is shown in blue, the $S_r A_r$ in green, the $S_i A_r$ in red and the $S_r A_i$ in black. h) MSD- Δt profiles of the experimental and the simulated replicas. The experimental replica is shown in blue, the $S_r A_r$ in green, the $S_i A_r$ in red and the $S_r A_i$ in black. The small letter i means intact whereas the small letter r means random.

Figure 4.6a-c. Randomized step lengths and directions of confined-like trajectories. a) Simulated confined-like trajectory. b) Three simulated trajectories obtained from the simulated trajectory shown in a. The simulation was done by keeping the order of step directions intact whereas the order of step lengths was randomized. c) Three simulated trajectories obtained from the simulated trajectory shown in a. The simulation was done by keeping the order of step lengths intact whereas the order of step directions was randomized.

Figure 4.6d-f. MSD plots of the trajectories shown in a-c. d) MSD- Δt plot (green) of the trajectory shown in figure 4.6a. The black line shows the theoretical MSD plot. e) MSD-dt plot of 100 simulated trajectories as described in the caption of figure 4.6b (shown in red). The green plot is the average of the 100 MSD plots. The MSD plots clearly indicate confined diffusion as the original trajectory shown in figure 4.6a. The black line shows the theoretical MSD plot. f) MSD- Δt plot of 100 simulated trajectories as described in the caption of figure 4.6b (shown in green). The MSD plots clearly indicate random diffusion. The black line shows the theoretical MSD plot.

Case 2:

To explain why the Hurst-correlation is lost by randomizing the angles and not by randomizing the displacements, we show—in figure 4.6g,h—the distribution of the diffusion coefficients and the angles of the 6.6kbp DNA molecules. As shown in these figures, the distribution of the diffusion coefficients (50 Δt time windows) follows a peak-type distribution (can be approximated to a Gaussian distribution). However, the distribution of the angles follows a random non-peak type distribution. In the experimental trajectories, the sub-distribution of long displacements (c-LO) and that of short displacements (d-HO) are matched with the angles that favor confined-like and directed motion, respectively (step-to-mode matching). Thus, the experimental Hurst profile (the blue plot in manuscript Figure 3g) shows significant deviation from the fully random simulation (the green plot). In the simulated trajectories with randomized displacements only,

the c-LO and the d-HO sub-distributions are lost because they are averaged during the randomization. As a result, the distribution becomes much narrower compared with the original one because the extreme (c-LO and d-HO) values average out to $D = 0.17 \mu\text{m}^2/\text{s}$ (Fig. 4.6g). Meanwhile, the step-to-mode matching is lost with loss of the deviation of the Hurst profile. On the other hand, in the simulated trajectories with randomized angles only, the c-LO and the d-HO sub-distributions are still intact. Before randomization of angles, we have three types of them within the $50\Delta t$ time scale: 1) angles that favors confined mode, 2) angles that favors directed mode and 3) angles that favor standard diffusion. Although the angles are randomized, the randomization —because of the statistical variations in the randomized angles— results in the same three types of the angles. Thus —again— there is a chance that some of the c-LO displacements match confined-like mode and some of the d-HO displacements match directed-like mode. Taken together, this discussion explains why the Hurst-correlation is lost by randomizing the displacement and not by randomizing the angles.

Figure 4.6g-h. Frequency distribution of the diffusion coefficients and the angles of 6.6kbp DNA. g) Frequency distribution of the diffusion coefficients. Each value was calculated based on a $50\Delta t$ time window. The red color encodes the original, experimental distribution, whereas the green color encodes the distribution after randomizing the order of displacements. h) Distribution of the angles in π scale.

As per the suggestion of the reviewer in this comment, we added the data of the nanoparticles in Figure S5.

Figure 3 and related text:

4.7) From what I understood, the authors took the experimental data (figure 1a: single, non-stitched time trace?) describing the mean position of the DNA as a function of time and fitted three different models of diffusion to a number of consecutive time points defined by a sliding window. The goodness of the fit then decides which model is most likely but no information is provided in whether the drift velocity or the positional accuracies have been fixed or not. Could the authors elaborate on their decisions? A free fit of the velocity, for example, could lead to situations where a low value of v makes a decision between a standard and directed diffusion rather arbitrary. On the other hand, fixing v to a certain value also seems arbitrary. Shouldn't be in any case the MSD (Δt) be calculated including the term for the localisation precision especially in hindsight which is here, as reasoned above, probably mainly coming from the non-Gaussian emission profile of the DNA?

We understand from this comment that reviewer raises a question about the fitting conditions of the time-dependent MSD plot to equation 10 ($MSD(\Delta t) = 4Dt + v^2(\Delta t)^2$) and about its reliability to distinguish between directed and standard diffusion. In order to respond properly to this important comment, we will consider a discussion with reverse order, i.e. we will start the discussion from the final MSD plots shown in the manuscript figure 7 & figure S7 and then end with equation 10.

The MSD profiles in the above mentioned figures show that the sub-trajectories which we obtained from the analysis showed considerable deviations from the standard diffusion. These deviations depict that the velocity (v) or the confined area side length (L) values which we obtained from the MSD plots showed considerable deviations from standard diffusion (manuscript figure 7c-d). To illustrate—in a statistical way— this issue, we show in figure 4.7a the statistical distribution of the v and L values. We show that our analytical approaches did not pick low v values or large L values that describe directed and confined motions deviating slightly from the standard diffusion.

To further illustrate—in a technical way— that the velocity or confined area values which we picked by the fitting do not correlate with the standard diffusion, we show—below—the manuscript figure 4c to illustrate how the c-LO and the d-HO events are picked. To choose the events that match confined-like motion associated with low occupancy or events that matches

directed-like motion associated with high occupancy, we superimposed the probability and MSD time traces and then applied the dynamic time warping algorithm (DTWA) to pick those events. After applying the DTWA, we only considered the peak values of the DTW profile (The values that showed great differences between the relative and absolute time traces). This corresponds to the 'o' marks in the below figure. Therefore, all values in the neighborhood ('x' mark in the below figure) were not considered. Because, we considered only the peak values (maximum deviations), the obtained v and L parameters after the fitting describe real directed and confined motions respectively.

In summary and based on the clarifications highlighted above, we only picked the trajectories that show significant difference between the probability and the MSD time profiles *i.e.*, when both profiles have an extreme difference value. Our final judgment on the mode of motion was based solely on the averaged MSD plots —not on the v and L values— when compared with the random simulation (manuscript figure S7a).

For the reviewer's technical comment on the fitting to equations 10, we note that we allowed free fit to the MSD plot to best describe the profiles and not to bias the fitting. For example, constraining the diffusion value (D) in equation S5 to $0.17 \mu\text{m}^2/\text{s}$ is inappropriate in this context because after we did the whole analysis we found that the tracks showed two D values (the displacement values of the c-LO is 80nm and that of the d-HO events is 54nm) rather than a single D value.

To respond to the comment on the positional accuracy, we note that the Gaussian kernel of the tracking algorithm fitted the fluorescence maxima —not the whole shape of the molecule— (Figure 4.1b). Once, the fitting to the Gaussian function fails, the tracking is terminated (Figure 4.1g). Therefore, the positions, indeed, resulted from Gaussian profiles with mean localization accuracy of 7.4 nm (Figure 4.3b). This accuracy value is approximately 10% of the displacement value of the 6.6kbp DNA molecule (66 nm). To compare this with the nanoparticles: their accuracy value (Figure 4.3a) is approximately 15% of their displacement value.

Figure 4.7 The distribution of the confined-area side length (L) and the speed (v) of the experimental replicas. The orange and the yellow zones highlight the range of the picked values of the c-LO and the d-HO sub-modes respectively. Extreme large L and extreme small v values were not considered in the analysis because our algorithms only picked the peak values. Therefore, the mode of motion of each sub-mode is not confused with standard diffusion.

Manuscript figure 4c. The 'o' marks show the events (greatest difference) we picked whereas the 'x' marks (less difference) show the events we did not consider.

4.8) Similarly, the authors decided to limit the L parameter necessary for describing confined diffusion to 700 nm (roughly four times the radius of gyration). The question remains how robust the analysis is or whether by tuning individual parameters the result of the MSD analysis would correlate with other modes of relative motion.

We hope that our evidences and discussions included in our response to comment 4.7 clarified that we are not looking at artifacts because of the fitting conditions. Our final conclusions remain solely dependent on the ensemble averaging data shown in manuscript figure 7 and not

on the time traces of the v and L values. In the context of our analysis, setting the limit of L at 700nm is equivalent to setting the standard diffusion itself as a limit to distinguish directed and confined motions. We found that when the fitting to equation 11 results in L value greater than 700nm, the MSD profile is better represented by standard diffusion rather than confined-like diffusion. Since equation 11 can also fit standard diffusion and even could result in infinite L value (Note that the L value obtained from the fitting is simply the asymptote of the MSD profile at ∞), we have to limit the fitting to a certain L value. Above this value, the mode is better described by standard or directed diffusion. Intuitively, It is not appropriate to classify the motion of a molecule translating inside large closed area (>700 nm in our analysis) within a short time scale ($50\Delta t$ in our analysis) as confined motion. We further note that we set the L value at 700 nm in all our random simulations and also during the analysis of the nanoparticles' trajectories (L limit of the nanoparticles = 400 nm). We did not find any false positive correlation in the motion for any of our simulations or for the nanoparticle tracks as shown in manuscript figure 6.

4.9) Furthermore, the authors classify four different sub-modes in figure 3 (c-LO, d-LO, d-HO, and c-HO), but continue to discuss mainly c-LO and d-HO. What happened to the other sub-modes? Are they here irrelevant?

For the d-LO and the c-HO sub-modes, we did not find any deviation in the experimental data compared with the random simulation (manuscript figure S7b).

4.10) Again, it would be interesting to see the comparisons between DNA and bead data diffusing under similar conditions.

Although we consider that our calculated Hurst exponent of the nanospheres at $3\Delta t$ is representative to that of DNA at $1\Delta t$ and is comparable, we agree with the reviewer that it is important to reproduce the data of the nanospheres at the same imaging conditions. We repeated the experiment and did our calculations on nanospheres imaged under the same experimental conditions. Because, the diffusion of the nanospheres is much faster in less glycerol concentrations ($D=1.3\mu\text{m}^2/\text{s}$), we could not obtain the same average tracking time of the

individual trajectories (The average tracking time of 6.6kbp DNA is $304\Delta t = 1.95\text{s}$ per trajectory, whereas the average tracking time for the nanospheres is $98\Delta t = 0.63\text{s}$). We did our analysis under this condition and reproduced the cumulative Hurst exponent graph. The plot (figures 4.10) shows identical profile to the imaged nanospheres at 70% glycerol (manuscript figure 3d), and this indicates pure random motion.

Figure 4.10. Hurst exponent (HE)- $i\Delta t$ profiles of nanospheres (purple) showing similar profile to manuscript figure 3d.

Additional comments:

4.11) It might be interesting at a later point to compare the results of the DNA labelled with Cy5 with data from DNA stained with an intercalating dye. One would expect that the observed time scales describing the modes of relative motion would shift especially if the density of intercalators is high.

We do thank the reviewer for suggesting this experiment and outlining the impact of intercalators on the speed of the relative dynamics. Because the labeling density of the intercalator dyes affects the elastic properties of the DNA, we agree with the reviewer that a shift in the time scale of the relative dynamics is expected. As recommended by the reviewer, we will consider this experiment in our experimental agenda. Meanwhile, we note that we do not use DNA intercalator dye, therefore the effect of the labeling on the elastic properties of DNA should be relatively small.

4.12) In the current analysis, any potential effects regarding 3D movement of the DNA, transient DNA-surface interactions, photo-physical effects of the Cy5 emitters (blinking, photo-bleaching, self-quenching,...) have not been taken into account. This simplification is likely to be valid, but highlights the difficulty on interpreting the results especially concerning the dynamics.

Regarding the 3D movement of the DNA, we addressed this issue in comment 3.3 and found that our observation is not related to the 3D motion of DNA.

Regarding the DNA-surface interactions, we were careful to image the molecules in the bulk of solution (roughly 50 μ m above the cover slip). Therefore, we exclude any surface-related interactions in our experiment.

Regarding the photo-physics of the Cy5 dye and their potential effect on the dynamic, we highlight that both the time and length scales of the relative dynamics are drastically different compared with the photophysical phenomena of organic dyes. Specifically, the length (0.44 μ m) and the time (0.33s) scales of the relative dynamics are several orders of magnitudes larger compared with the length and time scales of fluorescence quenching/energy transfer/blinking. The later fluctuations lie much below our temporal (6.4ms) and spatial (66 nm) resolutions. Also, we note that a mean of 300 dyes are attached to one molecule of DNA. Therefore, the fluctuations due to blinking and other effects should be averaged out in the final single molecule image. Furthermore, the negative results of the control experiment of Lambda DNA (comment 3.3) exclude the effect of the photophysical phenomena on the relative dynamics of the 6.6kbp DNA.

Reviewer #1 (Remarks to the Author):

The authors have significantly improved the presentation of their work, clarifying the abstract, and giving better explanation of the basic concepts of their approach. The advantages and disadvantages relative to CA tracking are now also highlighted.

We thank the positive comment of the reviewer.

However, I have a problem with their conclusion that friction anisotropy cannot explain the observed directed motion. The explanation hinges on their statement that the time scale of reorientation of the major direction of friction anisotropy (τ_{θ}) is much smaller than the conformational relaxation time (τ_R) of the molecule. I am not convinced that this must be the case. For a random coil (or long wormlike chain) I think these two time scales should be of the same order of magnitude. Evidence of this can be found in the authors' own supplementary video of DNA motion: instantaneously, the distribution of the fluorescent dye is clearly non-spherical (at the beginning of the video it has a longest axis somewhat along the vertical direction). This is what one can expect for a random coil or long wormlike chain. The question is how fast does this longest axis decay to isotropy or reorient? The video shows that during the time the center-of-mass (blue point) diffuses over a distance equal to the typical radius of the fluorescent distribution (I assume this distance is already larger than or at least equal to the radius of gyration of the DNA molecule), the longest axis has not changed appreciably. In other words, I think the authors in their explanation unjustly neglect the correlation between the anisotropic mass gyration tensor and the anisotropic center-of-mass diffusion tensor.

We thank the reviewers for the inspiring discussion on the origin of the relative sub-modes which we detected in DNA motion. We believe that our point of view and that of the reviewer are mutually supporting a positive correlation between the anisotropic diffusion tensor and the *d-HO* sub-mode (directed high occupancy sub-mode). However, as we included in the discussion section of the manuscript, we believe that the *d-LO* sub-mode is not solely related to this anisotropic diffusion. We argue this claim through the following discussion.

As we discussed in our responses to the reviewer, the anisotropic diffusion of semiflexible, long DNA can be modeled as the diffusion of elongated ellipsoid structures. The diffusion of these structures involves two components (D_a : diffusion in directions parallel to the long axis ($D_a = kBT/\gamma_a$) and D_b : diffusion in directions perpendicular to the long axis ($D_b = kBT/\gamma_b$)). As we argued, the molecule could show directed-like mode of diffusion (1D-like motion) during the D_a diffusion because of shape anisotropy. This 1D-like motion, however, should occur in both directions and with equal probabilities. Because of the stochastic nature of this 1D-like motion, one could observe that, sometimes, certain direction is transiently favored over the other direction. This, indeed, results in a transient directed-like motion because of shape anisotropy. However, the overall D_a diffusion should be regarded as random 1D-like motion. Therefore, the time scale of this transient directed-like motion—occurring during the D_a diffusion—should be less than the time scale of the whole conformational relaxation process.

In movie 1, we agree with the reviewer that this molecule stays in the relaxed conformational state for longer time. However, we do not have conclusive evidence that the time scale of the transient directed-like motion is comparable to the conformational relaxation time. To support this claim, the motion should be sampled at much faster time scale. Our camera frame rate (6.4ms) is only 1/8 the time required by molecule to diffuse a distance equal to the radius of gyration. Faster frame rates, however, are not accessible by using widefield microscopy. Because of this uncertainty and because of the fact that shape isotropy/anisotropy cannot similarly explain the *c-LO* sub-mode, we find that it is safer to conclude that anisotropic diffusion could explain the *d-HO* sub-mode—only partially—. Because of this argument, which is supported by the reviewer’s rational; we revised the text to further emphasize this possibility.

“Further to the above-mentioned rational for the mechanism of the relative dynamics, we believe these dynamics could be related – in part – to the anisotropic diffusion of DNA because of its transient relaxation. During the time when the DNA molecule is relaxed, its shape is anisotropic, which can be modelled as an elongated ellipsoid where a (length) $\gg b$ (width). The resulting anisotropic diffusion involves two components, D_a , diffusion coefficient in directions parallel to the long axis ($D_a = k_B T / \gamma_a$), and D_b , diffusion coefficient in directions perpendicular to the long axis ($D_b = k_B T / \gamma_b$). Because the friction coefficient γ_a is smaller than γ_b , D_a is greater than D_b and therefore the molecule is expected to show directed-like motion. The time scale of this directed-like motion is determined by τ_θ (the time required for the ellipsoid to diffuse 1 rad by rotational motion). At time scales longer than τ_θ , the rotation randomizes the motion and, eventually, results in a crossover from anisotropic diffusion to isotropic diffusion. Because the elongation of the DNA occurs transiently during the conformational relaxation process, we cannot gather conclusive evidence on whether τ_θ is correlated with τ_R and with τ_{c-LO} . Because of this uncertainty and the fact that shape isotropy/anisotropy cannot similarly explain the confined-like sub-mode motion, we argue that anisotropic diffusion could – only partially – explain the directed-like sub-mode motion. Taken together, we conclude that the conformational relaxation dynamics and the anisotropic diffusion partially elucidate the mechanism of the relative dynamics we describe here. Describing the full mechanism remains an open research question.”

Reviewer #3 (Remarks to the Author):

In this revised article, the authors newly performed measurement on lambda DNA as a control and compared two different localization methods for analyzing their images. These new measurement and analysis addressed my concerns before successfully. Thus, I recommend its publication in Nature communication. However, the readability of this paper has not been improved yet.

We greatly appreciate the positive comment of the reviewer. In the revised manuscript, we addressed the readability issue by adding Supplementary Schemes to the manuscript, which visually explain all the analyses used in this study step by step. Please see our reply to Reviewer #4 for the details.

Reviewer #4 (Remarks to the Author):

The revised manuscript "Conserved Linear Dynamics of Single Molecule (Brownian) Motion" has been considerably improved and my comments and suggestions have been carefully addressed. However, the manuscript is still very difficult to read and understand; the authors decision to move a significant part of the supplemental material into the main text did not help in that respect. My requested explanation of the Hurst coefficients, to give an example, introduced a new sentence which is difficult to understand: "Such fluctuations can be captured by detrending the temporal fluctuations by using detrended fluctuation analysis and by calculation of the Hurst exponent (HE)". I therefore strongly suggest another round focusing on readability and clarity.

We thank the reviewer for the suggestion. We agree with the reviewer that readers may have difficulty in understanding the analytical tools we use in the study that include detrended fluctuation analysis, Hurst exponent, and dynamic time warping algorithm. Although these are well established mathematical tools and all the details of the analyses are provided in the Methods section, the descriptions could be difficult to understand for general readership. Therefore, in the revised manuscript, we added step-by-step descriptions of all the analyses using schematic figures and illustrations (Supplementary Scheme 1-7). These Supplementary Schemes visually explain how all the analyses are conducted and thus complement the descriptions in the main text and the Methods section. We believe that these Supplementary Schemes greatly enhance the readability of the paper.

Localisation/Tracking: The added explanation on the chosen algorithm is helpful, especially as the authors compared the centre of mass approach with the finally used LTA (abbreviation was not introduced!) algorithm and found similar results.

We thank the reviewer for the positive feedback. We introduced the abbreviation LTA (localization and tracking algorithm) in our response to comment 3.2 in our previous reply letter. We did not use this abbreviation in the manuscript. The abbreviation was defined (in the Supplementary Note 1-A2) at first use.

The authors further decided to include data from lambda phage DNA, but could not show any diffusion sub modes as the increased radius of gyration shifts the crossover point quadratically to longer times, which cannot be accessed experimentally. Now we have the situation that we have two negative "controls" (lambda phage DNA and spheres), but only one positive control. Maybe the authors can estimate which length scales under which conditions they expect to show sub modes?

We thank the reviewer for pointing out this comment. We believe that our response to this comment would support the discussion on the applicability of our analysis under different conditions.

The time scale of the relative dynamics of 6kbp DNA which we captured is 0.33 s (52 Δt). Under our experimental condition, the average time-scale of the experimental trajectories is 1.95 s (304 Δt).

Therefore, to reliably capture the dynamics, we believe that the time scale of the experimental trajectories should be 3-6 times longer than the time scale of the relative dynamics.

To estimate the time scale needed to capture the relative dynamics of lambda DNA, if any, we would, first, need to extrapolate the length scale trend of the 6kbp DNA. The length scale of the c-LO sub-mode of the 6.6kbp molecule is $0.44 \mu\text{m}$ ($\approx 2.5R_g^{6kbpDNA}$). If we extrapolate this length-scale trend, we would need to capture a length scale of $1.75 \mu\text{m}$ ($\approx 2.5R_g^{\lambda DNA}$) for lambda DNA. This corresponds to a time scale of 3 s. Therefore, to capture the relative dynamics of lambda DNA, if any, we need 3-6 times longer time scale of the experimental trajectories, i.e., approximately 15 s ($2000 \Delta t$).

Therefore, we conclude that our analysis is limited by the time scale that widefield single molecule microscopy can capture. This time scale should be 3-6 times longer than the time scale of the relative dynamics that corresponds to length scale of $2.5R_g$.

We added the following sentences in the revised manuscript to describe the capability and limitation of our method.

“For the diffusion mode to be reliably captured using lattice occupancy analysis, the time scale of the dynamics should be slower than the frame rate of the detector and faster than the diffusion of the molecule out of the focal plane of the microscope. This limits the time scale of the dynamics that can be captured by lattice occupancy analysis. A detector with a faster frame rate and 3D single-molecule tracking techniques may further expand the applicability of the analysis to wider time scales.”

Point-by-point response to the referees

Reviewer 1

1. *The authors have dealt with my last remaining issue, regarding the correlation between anisotropic mass distribution and anisotropic diffusion and their relevant timescales, in a satisfactory manner. I agree with their conclusion. I have no further comments.*

Thank you for the comment. We are pleased that the reviewer is convinced by our interpretation of the results.

Reviewer 4

1. *The authors have again improved their manuscript (especially the readability) and I therefore recommend the manuscript for publication in Nature Communications.*

Thanks to the reviewer's constructive comments, we believe that the manuscript is now readable for broad *Nature Communications* readership.

2. *One more note, an effective time resolution considerably faster than the frame time of any camera can be achieved using stroboscopic laser excitation (<http://dx.doi.org/10.1039%2FC5CP04137F>).*

Thank you for the comment. Indeed, stroboscopic laser excitation could enhance effective time resolution of the imaging experiments. We cited the paper in the revised manuscript.